# Spatial localization of hippocampal replay requires dopamine signaling

Matthew R Kleinman*, David J Foster*

Helen Wills Neuroscience Institute and Department of Psychology, University of California, Berkeley, Berkeley, United States

## eLife Assessment

This work provides a **valuable** contribution to our understanding of the neurobiological mechanisms underlying spatial memory and learning, suggesting that dopamine plays a pivotal role in linking reward context and novelty to memory consolidation processes. The evidence presented to support the main conclusions is **solid**, although reviewers felt that the strength of evidence could have been further strengthened by more rigorous histological verification of the experimental conditions and the complexity of the experimental manipulations, increased sample sizes, and a more consistent approach to experimental dosing and timing, which will be crucial for confirming the reproducibility and reliability of the observed effects.

**\*For correspondence:**
mattrkleinman@berkeley.edu
(MRK);
davidfoster@berkeley.edu (DJF)

**Competing interest:** The authors declare that no competing interests exist.

## Abstract

Sequenced reactivations of hippocampal neurons called replays, concomitant with sharp-wave ripples in the local field potential, are critical for the consolidation of episodic memory, but whether replays depend on the brain's reward or novelty signals is unknown. Here, we combined chemogenetic silencing of dopamine neurons in ventral tegmental area (VTA) and simultaneous electrophysiological recordings in dorsal hippocampal CA1, in freely behaving male rats experiencing changes to reward magnitude and environmental novelty. Surprisingly, VTA silencing did not prevent ripple increases where reward was increased, but caused dramatic, aberrant ripple increases where reward was unchanged. These increases were associated with increased reverse-ordered replays. On familiar tracks this effect disappeared, and ripples tracked reward prediction error (RPE), indicating that non-VTA reward signals were sufficient to direct replay. Our results reveal a novel dependence of hippocampal replay on dopamine, and a role for a VTA-independent RPE signal that is reliable only in familiar environments.

## Introduction

Spatial information is encoded in the firing of hippocampal place cells, which are thought to provide a cognitive map to support memory and navigation (*O'Keefe and Dostrovsky, 1971*; *O'Keefe and Nadel, 1978*). During pauses in locomotion, place cells participate in structured population bursts of activity, representing temporally compressed trajectories through experienced locations, a phenomenon termed replay (*Diba and Buzsáki, 2007*; *Foster and Wilson, 2006*). Replay sequences can occur in the same order as experience, called forward replay, or in the reverse order of experience, called reverse replay (*Diba and Buzsáki, 2007*; *Foster and Wilson, 2006*). Replay appears after just one experience in a novel environment (*Berners-Lee et al., 2022*), and is preferentially generated toward goals in goal-directed tasks (*Pfeiffer and Foster, 2013*; *Widloski and Foster, 2022*). Experimentally interrupting or lengthening replay-associated sharp-wave ripples (SWR) in the local field potential (LFP) disrupts and enhances learning of a spatial memory task, respectively (*Fernández-Ruiz et al.,*

*2019*; *Jadhav et al., 2012*). Replay is thus thought to support memory consolidation and online planning (*Buzsáki, 2015*; *Foster, 2017*).

Intriguingly, reward drives increased rates of SWR (*Singer and Frank, 2009*), and only reverse replay, not forward, is increased at highly rewarding locations (*Ambrose et al., 2016*). Theoretical work suggests replay functions to update spatial representations of value to influence behavior and optimize reward receipt (*Mattar and Daw, 2018*). These findings hint that replay may be strongly modulated by reward-processing areas in the brain, such as the midbrain dopamine system (*Fields et al., 2007*). Dopamine neuron activity in the ventral tegmental area (VTA) is consistent with coding of reward prediction error (RPE), with increased activity at unexpected rewards and decreased activity with omission of expected rewards (*Schultz et al., 1997*). Subsequent work investigated dopamine release in spatial tasks, finding it ramps toward large rewards (*Guru et al., 2020*; *Howe et al., 2013*) in a manner consistent with encoding of RPE for a value function over space (*Kim et al., 2020*).

Besides the well-established role of midbrain dopamine neurons in reward processing, dopamine release in hippocampus has been implicated in stabilizing place fields (*Kentros et al., 2004*), gating the increase in plasticity in dorsal CA1 synapses by novel experiences (*Li et al., 2003*), and improving memory retention via increasing replay (*McNamara et al., 2014*). Furthermore, VTA activity is increased in novel environments (*Guru et al., 2020*; *McNamara et al., 2014*; *Takeuchi et al., 2016*), suggesting the hippocampus and VTA may coordinate to signal spatial novelty and induce learning in new environments (*Lisman and Grace, 2005*). However, recent work implicates locus coeruleus (LC) as the dominant source of dopaminergic input to dorsal CA1 and show its necessity and sufficiency for novelty-mediated episodic memory consolidation (*Kempadoo et al., 2016*; *Takeuchi et al., 2016*), leaving the role of VTA unclear.

We therefore tested whether VTA dopamine neurons are required for reward-related modulation of SWR and replay. We expressed an inhibitory DREADD (*Armbruster et al., 2007*) in VTA dopamine neurons and implanted a tetrode microdrive above hippocampus. We could then inhibit VTA dopamine signaling and simultaneously record neural activity in the dorsal CA1 region of hippocampus while rats collected rewards in familiar and novel environments. If VTA dopamine signaling is required for coordinating replay to valuable locations, we expected to see deficits in the capacity for reward to recruit SWR and replay. Additionally, if VTA dopamine is critical for inducing plasticity in CA1 in novel environments, novelty may significantly increase the effect of VTA inactivation on hippocampal replay.

## Results

### VTA inactivation in a simple spatial task with reward changes

We combined tetrode recordings in dorsal CA1 (dCA1) and chemogenetic silencing of VTA dopamine neurons to determine whether reward-related changes in hippocampal ripples and replay required VTA dopamine signaling. Transgenic rats expressing cre-recombinase under the tyrosine hydroxylase (TH) promoter were stereotactically injected with cre-dependent virus containing the inhibitory DREADD hM4Di (Experimental, $n = 4$) or mCherry-only control (Control, $n = 3$) into bilateral VTA (*Figure 1A*). We observed widespread expression across the extent of VTA and co-localization with TH (*Figure 1B*, *Figure 1—figure supplement 1*), enabling specific and reversible inactivation of VTA dopamine signaling. Post-experiment histology confirmed overlapping virus expression and TH-positive neurons in putative VTA near the injection site (–5.6 mm AP from bregma), as well as approximately 0.5 mm anterior and posterior (−5 to −6 mm AP). Recording microdrives containing 6–32 independently adjustable tetrodes (bilateral 32 tetrodes, $n = 4$; unilateral 20 tetrodes, $n = 2$; unilateral 6 tetrodes $n = 1$) were implanted above dCA1 (*Figure 1A*). Tetrodes were lowered to the pyramidal cell layer of dCA1, using the presence of SWR with upward deflections in the LFP, recording depth characteristic of dCA1, and spatially restricted firing of place cells to confirm the recording location.

Before each experimental session, rats were given intraperitoneal injection of CNO, to activate hM4Di receptors and suppress VTA dopamine neuron activity, or saline, then performed a simple task on linear tracks (1.5–2.5 m in length), collecting liquid chocolate rewards from each end (*Figure 1C*). Each session began with equal 0.1 ml reward volume at each end (Epoch 1) for 10–20 laps (1 lap was comprised of reward collection at both ends; mean, 16 laps). This was followed by unsignaled quadrupling of reward at one end to 0.4 ml (Incr. end), while reward at the other end remained unchanged

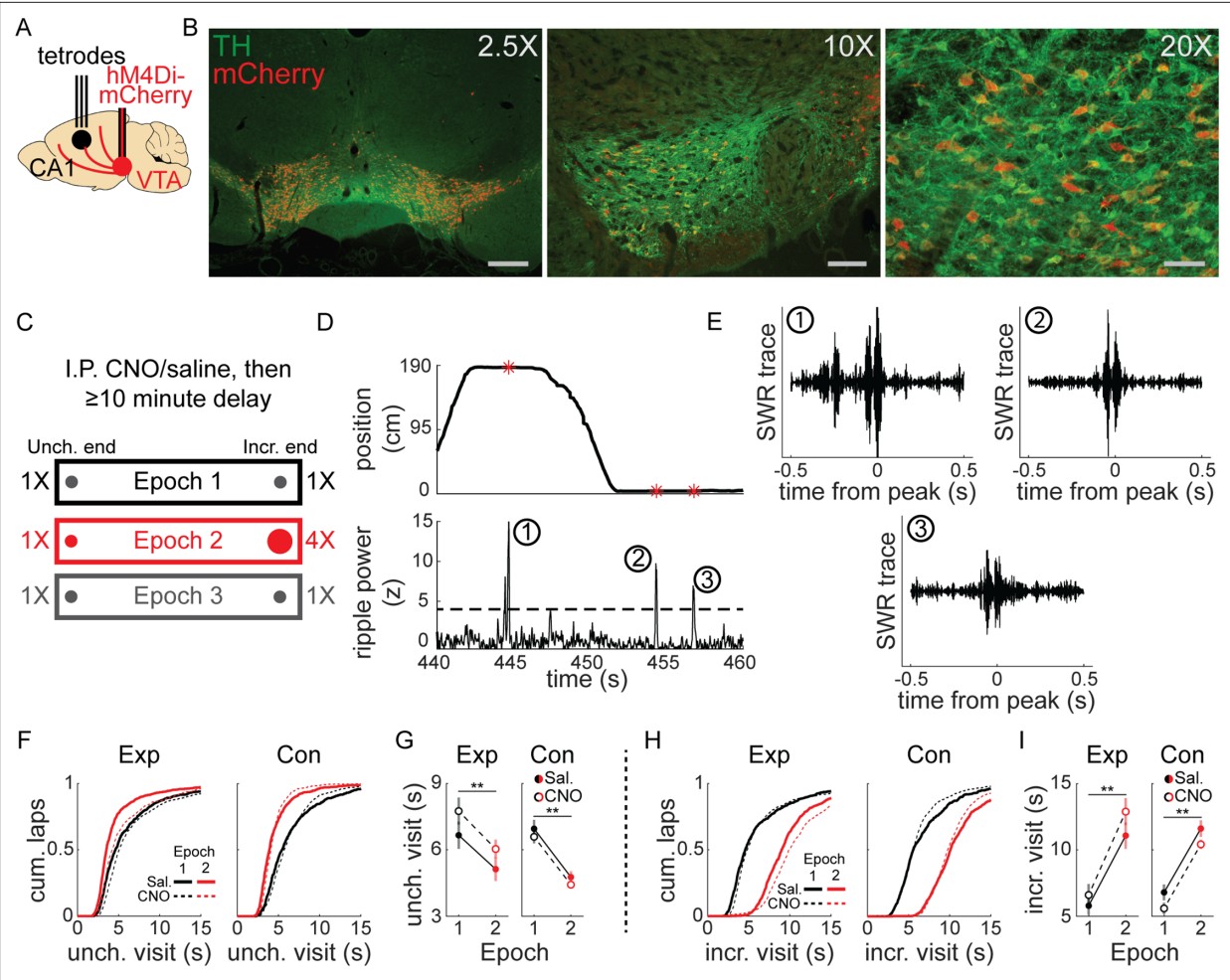

**Figure 1.** Experimental design and linear track behavior. (**A**) TH-cre rats underwent stereotactic surgery to inject virus bilaterally into ventral tegmental area (VTA) and implant a tetrode microdrive above dorsal CA1. (**B**) Co-expression of mCherry (red) and tyrosine hydroxylase (TH) (green) in VTA from three example animals. Left panel, mCherry-only virus, scale bar 600 µm; middle panel, hM4Di-mCherry, scale bar 150 µm; right panel, hM4Di-mCherry, scale bar 75 µm. (**C**) Intraperitoneal injection of saline or CNO (1–4 mg/kg) preceded recording sessions by at least 10 min. Rats were placed at one end of a linear track and collected liquid chocolate reward from wells at each end. Each epoch lasted 10–20 laps and reward changes were unsignaled to the animal. For each session, the Incr. end was defined as the reward end with 4X reward in Epoch 2, and the Unch. end was defined as the reward end with 1X reward in Epoch 2. (**D**) During stopping periods at reward ends, local field potential (LFP) was band-pass filtered in the ripple band (150–250 hz) and sharp-wave ripples (SWR) events were detected. (**E**) Three example ripple-filtered LFP traces from one lap (two stopping periods) are shown. (**F**) Cumulative distribution of reward end stopping periods at the Unch. reward end in Epoch 1 and 2 for experimental rats (left panel) and control rats (right panel). See also *Figure 1—figure supplement 2*. (**G**) The duration of Unch. reward end stopping periods decreased from Epoch 1 to Epoch 2. The mean stopping durations in each condition were calculated per session, then analyses performed across sessions. Mean ± standard error, Exp Saline, Epoch 1: 6.67 ± 0.61, Epoch 2: 5.13 ± 0.53; Exp CNO, Epoch 1: 7.766 ± 0.61, Epoch 2: 6.04 ± 0.42. Con Saline, Epoch 1: 6.96 ± 0.4, Epoch 2: 4.77 ± 0.27; Con CNO, Epoch 1: 6.59 ± 0.29, Epoch 2: 4.43 ± 0.15. Mixed-effects generalized linear model (GLM) with epoch, drug, animal group, and all interactions, with individual animal as a random effect: epoch, $z = -3.37$, $p < 0.001$; all other terms, n.s. Total session count: experimental group, n = 45 saline and n = 44 CNO; control group, n = 35 saline and n = 37 CNO. (**H**) Cumulative distribution of reward end stopping periods at the Incr. reward end in Epoch 1 and 2 for experimental rats (left panel) and control rats (right panel). (**I**) The duration of Incr. reward end stopping periods increased from Epoch 1 to Epoch 2. The mean stopping durations in each condition were calculated per session, then analyses performed across sessions. Mean ± standard error, Exp Saline, Epoch 1: 6.64 ± 0.54, Epoch 2: 10.87 ± 0.8; Exp CNO, Epoch 1: 7.28 ± 0.66, Epoch 2: 12.31 ± 0.82. Con Saline, Epoch 1: 7.44 ± 0.78, Epoch 2: 11.29 ± 0.5; Con CNO, Epoch 1: 6.47 ± 0.35, Epoch 2: 10.34 ± 0.27. Mixed-effects GLM with epoch, drug, animal group, and all interactions, with individual animal as a random effect: epoch, $z = 4.62$, $p < 10^{-5}$; all other terms, n.s. *$p < 0.05$; **$p < 0.01$.

The online version of this article includes the following figure supplement(s) for figure 1:

**Figure supplement 1.** Additional histology examples.

**Figure supplement 2.** Behavioral effects of novelty and ventral tegmental area (VTA) inactivation.

**Figure supplement 3.** Effect of reward change on running velocity.

(Unch. end), for 10–20 laps (Epoch 2; mean, 16.7 laps). Finally, reward was equalized again to 0.1 ml at both ends (Epoch 3) for up to 20 laps (mean, 11.6 laps).

Each animal performed this task on familiar linear tracks (>2 sessions on track; total session count for each condition: Experimental rats: 36 saline, 34 CNO; Control rats: 23 saline, 25 CNO) and novel linear tracks (first or second session on track; Experimental rats: 9 saline, 10 CNO; Control rats: 12 saline, 12 CNO). During stopping periods at either end of the track (velocity ≤8 cm/s, position ≤10 cm from end), SWR were identified as peaks in the ripple band (150–250 hz) in LFP (*Figure 1D, E*; see Methods).

Gross behavior was largely unaffected by VTA suppression (e.g., all reward consumed on each lap), but CNO in experimental animals systematically affected stopping period duration. Visits to the Unch. end in Epoch 2 were significantly shorter than in Epoch 1, despite unchanged reward volume there, and this reduction was present in both saline and CNO sessions (*Figure 1F, G*). CNO did not affect control animals (*Figure 1F, G*). Visits to the Incr. end were significantly longer in Epoch 2 than in Epoch 1 in all conditions, owing to the increased reward consumption time (*Figure 1H, I*). Changes in stopping period duration in Epoch 3 were similar across all conditions: Unch. visit duration increased from Epoch 2 to Epoch 3 (*Figure 1—figure supplement 2E*), while Incr. visit duration decreased (*Figure 1—figure supplement 2F*). Separate analysis of novel and familiar sessions revealed the pattern of shorter duration Unch. visits in Epoch 2 compared to Epoch 1 did not depend on novelty (*Figure 1—figure supplement 2A–D*). However, the main effect of CNO in experimental rats of prolonging stopping periods occurred in novel sessions (*Figure 1—figure supplement 2A, B*), not in familiar sessions (*Figure 1—figure supplement 2C, D*). Rats consistently ran slightly faster toward the Incr. end than the Unch. end in Epoch 2, across all conditions (*Figure 1—figure supplement 3*).

We interpret the reduction in visit duration as a behavioral signature of the Unch. end becoming relatively less valuable during Epoch 2, when the reward volume is larger at the Incr. end. Though visit durations were slightly longer in CNO sessions in experimental rats, particularly in novel track sessions, this behavioral effect of a relative value decrease remained, indicating VTA inactivation did not prevent rats from recognizing a devalued location.

## Reward-related modulation of SWR rate is mediated by novelty and VTA

We analyzed the rate of SWR occurrence (number of SWR/total duration rat was stationary) during the first 10 s of each stopping period, when rats were consuming reward. In individual sessions, SWR rate increased robustly in all conditions shortly after stopping at the reward wells and beginning reward consumption (*Figure 2A*). Relative to Epoch 1, SWR rate during Epoch 2 tended to increase dramatically at Incr. end visits and decrease at Unch. end visits. During Epoch 3, SWR rate at the Incr. end dropped precipitously relative to Epoch 2, while often increasing at the Unch. end (*Figure 2—figure supplement 1A*).

Surprisingly, during novel sessions, VTA inactivation often led to increased SWR rate at both reward ends (*Figure 2A*, right). SWR rate still increased in Epoch 2 at the Incr. end even without normal VTA signaling, indicating reward sensitivity per se was not abolished, but suggesting the localization of this increased SWR rate to where reward increased was disrupted.

Pooling across sessions revealed this dramatic increase in SWR rate at the Unch. end was typical with CNO in novel sessions, and further suggested there was a reduction in the difference in SWR rate between the Incr. and Unch. ends in Epoch 2 in both familiar and novel experiences (*Figure 2B*). We therefore used a Poisson generalized linear model (GLM) to quantify the changes in SWR rate across reward end, epoch, drug condition, and novelty (see Methods). In experimental rats, CNO and reward both influenced SWR rate, with significant effects for the CNO main effect ($z = 3.19$, $p < 0.01$), the interaction between Incr. end and Epoch 2 ($z = 9.02$, $p < 10^{-10}$) and the three-way interaction between Incr. end, Epoch 2, and CNO ($z = -2.06$, $p < 0.05$). Control animals showed no apparent effect of CNO (*Figure 2C*). The same Poisson GLM fit to control rat data confirmed this, with significant coefficients only for Incr. end ($z = -2.42$, $p < 0.05$) and the interaction between Incr. end and Epoch 2 ($z = 7.64$, $p < 10^{-10}$). SWR duration was reduced in novel sessions, consistent with replays becoming longer with increased familiarity (*Berners-Lee et al., 2021*), as well as in Epoch 2, but was otherwise unaffected by reward or drug (*Figure 2—figure supplement 4*).

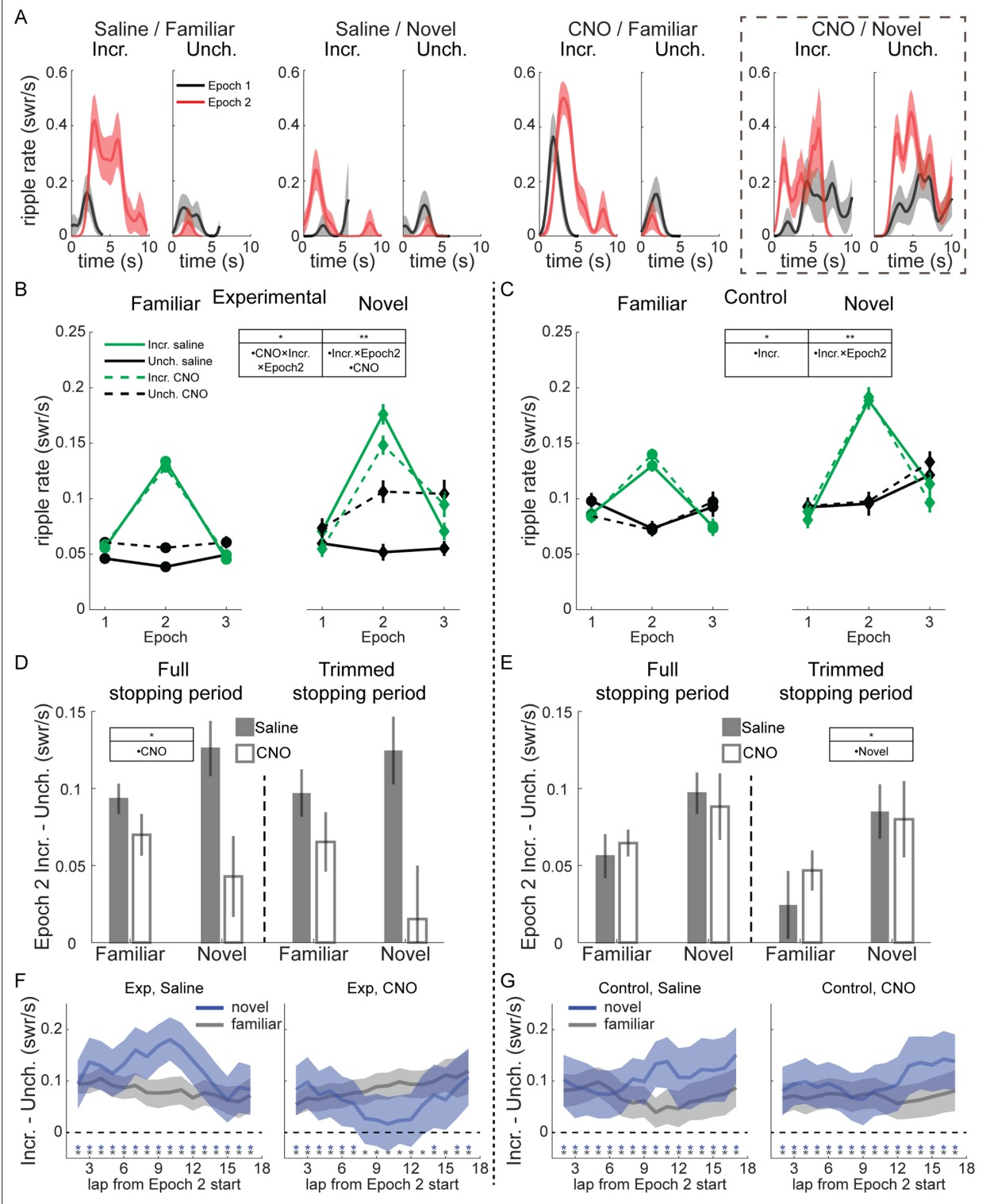

**Figure 2.** Modulation of sharp-wave ripples (SWR) rate by reward, novelty, and ventral tegmental area (VTA) inactivation. (**A**) SWR rate as a function of time in stopping period in Epoch 1 and 2 for four example sessions in experimental rats; from left to right, saline on familiar track, saline on novel track, CNO on familiar track, and CNO on novel track. In each panel, visits to the Incr. end are on the left and visits to the Unch. end are on the right. Relative to Epoch 1 (black lines), in Epoch 2 (red lines) SWR rate increased at Incr. end and decreased at Unch. end in all conditions except for CNO on a novel track (far right), where SWR rate increased at both ends in Epoch 2. SWR rate was binned in 0.25 s windows and smoothed with a 2-bin Gaussian. Line, mean; shading, standard error. (**B**) SWR rate in experimental rats as a function of epoch, drug (saline in solid lines, CNO in dashed lines), reward end (Unch. in black, Incr. in green), and novelty (familiar in left panel, novel in right panel). A mixed-effects Poisson generalized linear model (GLM) was fit

*Figure 2 continued on next page*

*Figure 2 continued*

to predict changes in SWR rate across reward end, epoch, drug condition, and novelty, with animal identity as a random effect. Significant coefficients: CNO ($z$ = 3.19, p < 0.01), the two-way interaction Incr. end × Epoch 2 ($z$ = 9.02, p < $10^{-10}$), and the three-way interaction between Incr. end × Epoch 2 × CNO ($z$ = −2.06, p < 0.05). *p < 0.05; **p < 0.01. (**C**) SWR rate in control rats as a function of epoch, drug (saline in solid lines, CNO in dashed lines), reward end (Unch. in black, Incr. in green), and novelty (familiar in left panel, novel in right panel). The same Poisson GLM fit to control rat data had significant coefficients: Incr. end ($z$ = −2.42, p < 0.05) and the two-way interaction Incr. end × Epoch 2 ($z$ = 7.64, p < $10^{-10}$). (**D**) Difference between SWR rate at Incr. and Unch. ends in Epoch 2 in experimental rats. Full stopping period, left panel. Trimmed stopping period, with first 1 s and last 1 s of visit excluded to eliminate all slow approaching/leaving movement, right panel. Saline, gray bars; CNO, white bars. Mean and standard error. A mixed-effects model with drug, novelty, and their interaction, and animal-specific intercepts (full model) was compared to reduced model lacking the drug terms (reduced model). Full stopping periods: full model, intercept, $z$ = 6.05, p < $10^{-5}$; other terms, n.s. $AIC_{reduced}$–$AIC_{full}$ = 5.22. Trimmed stopping periods: full model, intercept, $z$ = 3.5, p < 0.001; other terms, n.s. $AIC_{reduced}$–$AIC_{full}$ = 4.55. Total session count: experimental group, familiar, n = 36 saline and n = 34 CNO; experimental group, novel, n = 9 saline and n = 10 CNO. (**E**) Difference between SWR rate at Incr. and Unch. ends in Epoch 2 in control rats, as in (**D**). Full stopping periods: full model, intercept, $z$ = 3.95, p < $10^{-5}$; other terms, n.s. $AIC_{reduced}$–$AIC_{full}$ = −3.54. Trimmed stopping periods: full model, novelty, $z$ = 2.31, p < 0.05; other terms, n.s. $AIC_{reduced}$–$AIC_{full}$ = −3.1. Total session count: control group, familiar, n = 23 saline and n = 25 CNO; control group, novel, n = 12 saline and n = 12 CNO. (**F**) In experimental rats, the difference in SWR rates at each reward end (Incr. – Unch.) in Epoch 2, after subtracting the mean rates in Epoch 1, averaged over a 5-lap sliding window within Epoch 2. Blue lines, novel sessions. Gray lines, familiar sessions. Blue and gray asterisks denote the centers of sliding windows in which the difference in SWR rate was significantly greater than 0 in novel and familiar sessions, respectively (one-sample *t*-test, p < 0.05, uncorrected for multiple comparisons). Shading denotes 95% confidence interval. See also *Figure 2—figure supplement 1*. (**G**) As in (**F**), but for control animals.

The online version of this article includes the following figure supplement(s) for figure 2:

**Figure supplement 1.** Modulation of sharp-wave ripples (SWR) rate by reward increase.

**Figure supplement 2.** Sharp-wave ripples (SWR) rate in Epoch 3.

**Figure supplement 3.** Similar results independent of session number of the day.

**Figure supplement 4.** Ripple duration is increased with familiarity.

To assess the interaction between VTA inactivation and novelty, we fit the Poisson GLM separately to novel and familiar sessions, then used bootstrapping to generate distributions of SWR rates for each reward end and condition under the null hypothesis that CNO had no effect (see Methods). We found the actual difference in SWR rates between saline and CNO sessions in experimental animals during Epoch 2 was significantly greater than chance (*Figure 2—figure supplement 1A*) in novel sessions for both the Unch. end (one-tailed test, CNO > saline, p < 0.001) and the Incr. end (one-tailed test, saline > CNO, p < 0.01), as well as at the Unch. end in familiar sessions (one-tailed test, CNO > saline, p < 0.01). There was no significant difference between saline and CNO in control animals at either reward end in either familiar or novel sessions (*Figure 2—figure supplement 1B*; one-tailed tests, all n.s.).

A potential functional role for the reward-related changes in SWR rate is to strengthen downstream representations of particularly rewarding locations at the expense of less rewarding locations. We tested whether VTA suppression blunted the SWR rate difference between reward ends in Epoch 2. Across both familiar and novel environments, CNO reduced the difference in SWR rate between the Incr. end and Unch. end in experimental rats but not in control rats (*Figure 2D, E*, left panels; experimental group, full mixed-effects model with drug terms strongly outperformed reduced model lacking drug terms, $AIC_{reduced}$–$AIC_{full}$ = 5.22; control group, reduced model lacking drug terms strongly outperformed the full model, $AIC_{reduced}$–$AIC_{full}$ = −3.54). We saw little difference in this measure when comparing sessions that occurred first in each day to those that occurred later (*Figure 2—figure supplement 3*), and little behavioral evidence from running velocity or number of laps completed that motivation systematically varied across session number. To control for the possibility that VTA inactivation caused changes in locomotor or other non-consummatory behavior at reward wells that might affect SWR emission, we omitted the first and last 1 s of each stopping period to isolate the reward consumption period. The effect of CNO remained in experimental rats, reducing SWR rate discrimination between Incr. and Unch. ends (*Figure 2D, E*, right panels; experimental group, full mixed-effects model with drug terms strongly outperformed reduced model lacking drug terms, $AIC_{reduced}$–$AIC_{full}$ = 4.55; control group, reduced model lacking drug terms strongly outperformed the full model, $AIC_{reduced}$–$AIC_{full}$ = −3.1).

We next looked for within-epoch changes in SWR rate to determine whether VTA inactivation altered the dynamics of the response to reward changes. We calculated the difference in SWR rate

at the Incr. and Unch. reward ends (each with its Epoch 1 mean subtracted) in a 5-lap sliding window. In all time windows and conditions except novel CNO sessions in experimental rats, the SWR rate at the Incr. end was significantly greater than at the Unch. end (*Figure 2F, G*; Incr. – Unch. significantly greater than 0, one-sample *t*-test, p < 0.05, uncorrected for multiple comparisons).

In novel sessions, VTA inactivation did not prevent an initially larger increase in SWR rate at the Incr. end than the Unch. end, but caused that difference to diminish over laps (*Figure 2F*). By the middle of the epoch, there was no statistically significant difference in reward modulation between the reward ends (one-sample *t*-test, p > 0.05, for 5-lap windows centered on laps 8–13 and 15), consistent with an initial appropriately localized reaction to reward change that eventually spread across the track. We found a similar deficit in Epoch 3, with SWR rate decreasing significantly more compared to Epoch 2 at the Incr. end than the Unch. end (Incr. – Unch. significantly below 0, one-sample *t*-test, p < 0.05, uncorrected for multiple comparisons) for almost every task condition and timepoint except in novel CNO sessions in experimental rats (*Figure 2—figure supplement 2B, C*). This suggests VTA inactivation may disrupt the normal magnitude or time course of the SWR response to negative value changes as well.

Overall, VTA inactivation spared the capacity for increased reward to modulate SWR rate but led to decreased differentiation of low and high value locations, particularly in novel environments where SWR rate increased spatially indiscriminately.

## SWR rate is correlated with RPE even with VTA inactivation

Taken together, the above results demonstrate VTA inactivation caused changes in the normal dynamics of the response of SWR rate to positive and negative changes in reward value (*Figure 2*). However, because each session had only two timepoints when reward value changed by fixed amounts, our experiment was not optimized to probe the precise relationship between SWR rate and reward changes. Additionally, the effect of VTA inactivation was particularly prominent with novelty, when both SWR rate and its modulation by reward changes were greater, raising the possibility that large SWR rates and fluctuations, rather than novelty per se, depend on VTA dopamine signaling.

To address these questions, we designed a volatile reward schedule (Experiment 2) with frequent, large reward changes at one end of the linear track, and tested whether VTA inactivation impacted the capacity for SWR rate to track value (*Figure 3A*, top). The 'stable end' delivered 0.2 ml every lap, while the 'volatile end' reward volume was drawn pseudorandomly from 0, 0.1, 0.2, 0.4, and 0.8 ml (mean 0.37 ml; blocks of 20 laps were comprised of 3 laps × 0 ml, 4 laps × 0.1 ml, 3 laps × 0.2 ml, 4 laps × 0.4 ml, and 6 laps × 0.8 ml). The position of the stable and volatile ends randomly varied across sessions.

This reward schedule also allowed us to test whether SWR rate was correlated with value, RPE, or neither. We expected SWR rate at the volatile end would be predominantly determined by the current reward volume there, but potentially also modulated by previous reward volumes (*Figure 3A*, bottom). If SWR rate is correlated with value, then for a given current volume, larger reward volumes at the last visit will lead to higher SWR rates compared to when the last visit was a smaller reward volume. Conversely, if SWR rate is correlated with RPE, the opposite modulation by last reward volume will be observed: the larger the previous reward volume, the lower the current SWR rate.

A subset of rats performed sessions of the modified reward schedule (mean 53.4 laps per session; total sessions per condition: Experimental rats 2 and 4: 6 saline, 7 CNO; Control rats 1–3: 16 saline, 18 CNO). Each rat completed one to two saline sessions before any CNO sessions and all sessions were on the same linear track, meaning almost all CNO sessions took place on a familiar track. As expected, SWR rate at the volatile end was predominantly determined by the current volume (*Figure 3B, C*). There was little obvious difference between saline and CNO in either experimental (*Figure 3B*) or control rats (*Figure 3C*). SWR rate at the stable end was largely stable across laps, although there was a trend toward higher SWR rate if the most recent volatile end visit was lower volume, consistent with lap-by-lap changes in the relative value of the stable end (*Figure 3—figure supplement 1*).

We next investigated how SWR rate at the volatile end varied as a function of both current and immediately previous volatile end volume in experimental rats (*Figure 3D, E*, top panels). For each current volume, we subtracted the mean SWR rate across all previous volumes, and examined how previous volume affected the mean-subtracted SWR rates across all current volumes (*Figure 3D, E*, middle panels).

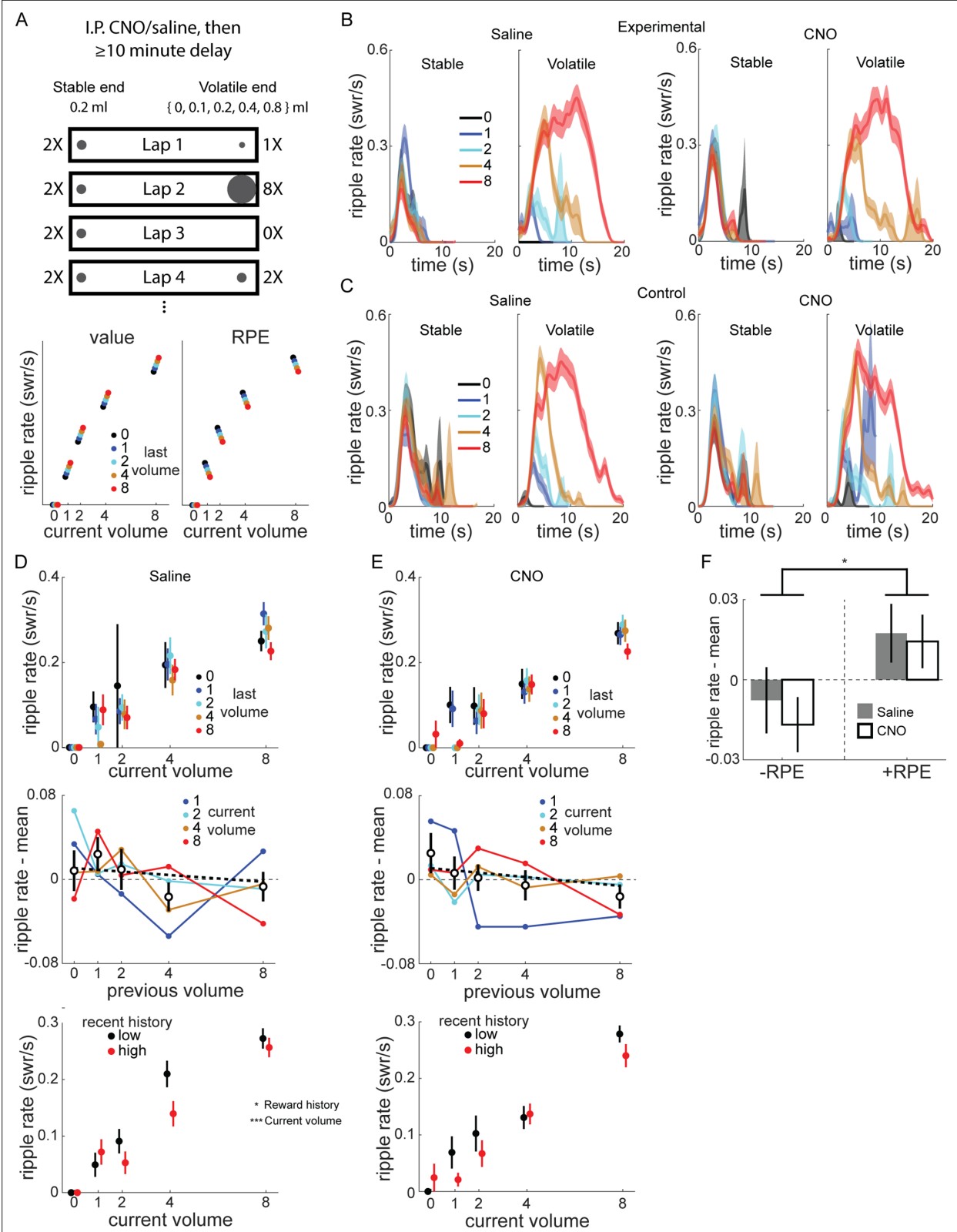

**Figure 3.** Frequent reward changes modulated sharp-wave ripples (SWR) rate. (**A**) Recording sessions in the volatile reward task were preceded by intraperitoneal injection of saline or CNO by at least 10 min. Rats were placed on the stable end to begin each session, which delivered 0.2 ml reward at each visit, while the volatile end delivered 0, 0.1, 0.2, 0.4, or 0.8 ml, pseudorandomly chosen on each lap. Bottom panel, schematic of how value and reward prediction error (RPE) would modulate SWR. Given a particular current volume, value coding predicts a positive correlation between SWR rate

*Figure 3 continued on next page*

*Figure 3 continued*

and previous volume, while RPE coding predicts a negative correlation. (**B**) SWR rate as a function of reward volume and time in end visit in example rat, Experimental rat 4. Left panel, saline. Right panel, CNO. In stable panel, traces are colored based on previous volatile end visit volume. In volatile panel, traces are colored based on current volatile volume. See also *Figure 3—figure supplements 1 and 2*. (**C**) SWR rate as a function of reward volume and time in end visit in example Control rat 3, as in (**B**). (**D**) Top panel, SWR rate at volatile end as a function of current and previous volatile volume, for saline sessions in experimental rats. Middle panel, SWR rate for each non-zero volatile volume plotted as a function of previous volume, with the mean SWR rate for that current volume subtracted. Unfilled symbols, mean of previous volume across all current volumes. Thick dashed line, linear fit to mean values. Pearson correlation between (ripple rate – mean) and previous volume, $r = -0.076$, $p = 0.177$. Error bars, standard error. Bottom panel, SWR rate as a function of reward volume, separated by recent reward history (median split on average of last three visits). Black, recent history below median; red, recent history above median. (**E**) Same as (**D**), for CNO sessions in experimental rats. Middle panel, Pearson correlation between (ripple rate – mean) and previous volume, $r = -0.109$, $p < 0.05$. Generalized linear model (GLM) fitting SWR rate as a function of drug, current volume, and previous volume: previous volume, $z = -2.31$, $p < 0.05$; drug and current volume, both $p > 0.8$. Bottom panel, Poisson GLM fitting ripple rate as a function of volume, drug condition, and reward history (above/below median), with animal-specific intercept as random effect: volume, $z = 13.86$, $p < 10^{-10}$; history, $z = -2.23$, $p < 0.05$; drug, $z = -1.05$, $p = 0.29$. (**F**) The RPE of volatile end visits were calculated by subtracting the previous volatile volume from the current volume. Two-way ANOVA with drug and RPE sign (+/−): drug ($F[1,518] = 0.3$, $p = 0.582$), RPE sign ($F[1,518] = 6.42$, $p < 0.05$), and drug × RPE sign ($F[1,518] = 0.07$, $p = 0.785$). *$p < 0.05$, ***$p < 0.001$.

The online version of this article includes the following figure supplement(s) for figure 3:

**Figure supplement 1.** Sharp-wave ripples (SWR) rate at stable end in experimental rats.

**Figure supplement 2.** Sharp-wave ripples (SWR) rate in all sessions in volatile reward task.

The mean-subtracted SWR rate was modestly negatively correlated with the previous volume (Pearson correlation: saline, $r = -0.076$, $p = 0.177$; CNO, $r = -0.109$, $p < 0.05$). A GLM found the mean-subtracted SWR rate was significantly affected by previous volume ($z = -2.3$, $p < 0.05$), but not drug ($z = -0.24$, $p = 0.81$) or current volume ($z = 0.196$, $p = 0.844$). There was no similar behavioral effect: for each current volume, we subtracted the mean reward end visit duration, and found no correlation between the previous reward volume and mean-subtracted visit duration (Pearson correlation; saline, $r = 0.011$, $p = 0.844$; CNO, $r = -0.075$, $p = 0.175$).

We next separated visits to the volatile end using a median split based on the recent volumes (mean of previous three visits). SWR rates were higher for a given current reward volume when the recent reward history was low (*Figure 3D, E*, bottom panels). A Poisson GLM predicting SWR rate as a function of current volume, drug condition, and whether reward history was low or high (relative to the median for the session) revealed significant effects of current volume ($z = 13.86$, $p < 10^{-10}$), as expected, and reward history split ($z = -2.23$, $p < 0.05$), but not drug condition ($z = -1.05$, $p = 0.29$).

Finally, we separated combinations of current and previous volume into those with negative RPE (current < previous) and positive RPE (current > previous) and found mean-subtracted SWR rate was significantly affected by RPE sign (*Figure 3F*; two-way ANOVA with drug and RPE sign, RPE sign: $F[1,518] = 6.42$, $p < 0.05$), but not drug ($F[1,518] = 0.3$, $p = 0.582$; drug × RPE sign, $F[1,518] = 0.07$, $p = 0.785$).

Given the lack of an effect of drug in experimental rats, we pooled all sessions (both animal groups, both drug conditions) in Experiment 2 to maximize experimental power and found similar results as in just experimental rats (*Figure 3—figure supplement 2*). On top of a large increase in SWR rate with current volume at the volatile end (*Figure 3—figure supplement 2A, B*), SWR rate was also significantly negatively correlated with the previous volatile volume, both at the volatile end (*Figure 3—figure supplement 2C*) and at the stable end (*Figure 3—figure supplement 2E*). Accordingly, SWR rates were significantly lower for negative RPE than for positive/non-negative RPE (*Figure 3—figure supplement 2D–F*). Finally, recent reward history at the volatile end significantly affected SWR rate (*Figure 3—figure supplement 2G*), with higher SWR rate when recent rewards were lower.

Taken together, SWR rate was modulated by reward volume changes consistent with RPE-like coding. This modulation did not require normal VTA dopamine signaling, at least in familiar environments. The lack of effect of VTA inactivation, even with frequent, large swings in value and SWR rate, corroborates the results from Experiment 1 that novel experiences are particularly susceptible to disruption, indicating VTA dopamine release is critical when learning new reward locations.

## Rate of reverse replay is increased with reward in novel environments only with intact VTA signaling

Previous work discovered the incidence rate of reverse replay, but not forward replay, was increased at locations with increased reward (*Ambrose et al., 2016*). We therefore analyzed single-unit data

collected in Experiment 1 (excluding Experimental rat 2, who had a 6-tetrode recording drive) to determine whether this modulation of replay required VTA dopamine signaling. As previously observed (e.g., *Ambrose et al., 2016*), place cells in dCA1 had directional fields, such that the location a neuron was active while the rat moved in one direction on the track (e.g., 'upward') was often distinct from its activity when the rat moved in the other direction (e.g., 'downward'). This directionality was apparent in both familiar and novel sessions, including in experimental rats with either saline or CNO (*Figure 4A*). We found no effect of CNO on within-session field reliability, but significantly less reliability in novel compared to familiar sessions (*Figure 4—figure supplement 1A*). Field similarity across running directions was slightly but significantly increased by both CNO and novelty (*Figure 4— figure supplement 1B*).

Two place fields were defined for each neuron, one for each running direction, permitting Bayesian decoding methods to estimate both position and direction from neural activity (*Figure 4B, C*). Sessions with accurate position and direction decoding during run, primarily due to sufficiently high cell yield (cell count, mean ± SEM; included sessions: 26.1 ± 1.1, excluded sessions: 9.5 ± 1.6; two-sample $t$-test, $t(133) = 5.3$, $p < 10^{-5}$) and also marginally smaller average place field size (mean place field size, defined as number of spatial bins with >1 Hz firing rate, mean ± SEM: included sessions: 47.7 ± 1.3, excluded sessions: 57.7 ± 5.8; two-sample $t$-test, $t(133) = -2.33$, $p < 0.05$), were included for replay analysis (*Figure 4—figure supplement 2*; total sessions included: Experimental rats: novel saline, $n = 8$; novel CNO, $n = 8$; familiar saline, $n = 18$; familiar CNO, $n = 23$; Control rats: novel saline, $n = 12$; novel CNO, $n = 11$; familiar saline, $n = 16$; familiar CNO, $n = 17$).

Candidate replay events were periods of high population spiking activity while the rat was not running ($z$-scored spike count >3, minimum duration of 50 ms, rat velocity ≤8 cm/s). We used a memory-less Bayesian decoder, with 40 ms decoding windows advancing by 5 ms steps, to estimate position and direction from neural activity. Replays were defined as candidate events with spatial trajectories meeting a threshold for motion and minimum total posterior in one running direction map (*Ambrose et al., 2016*) (see Methods), with the running direction with greater posterior probability used to classify replay directionality, described below.

Forward replays were spatial trajectories moving across the track in the same direction as the rat when fields were calculated, for example, moving 'downward' with posterior probability in the 'downward' place field map (*Figure 4B*, left panel). Reverse replays were trajectories that moved in the opposite direction as the rat, for example, moving 'upward' with posterior probability in the 'downward' place field map or vice versa (*Figure 4B*, middle and right panels). We found forward and reverse replays occurred in all conditions, including in novel sessions with saline (*Figure 4B*) or CNO (*Figure 4C*). We therefore asked how novelty and drug condition influenced the effect of reward change on rates of reverse and forward replay.

Consistent with previous work (*Ambrose et al., 2016*), the rate of reverse replay was strongly modulated by the reward volumes on the track. Excluding novel CNO sessions for the moment, in all other conditions in experimental rats, when reward was larger at the Incr. end than the Unch. end (unequal reward), reverse replay was significantly increased at the Incr. end relative to when rewards were equal (novel saline: equal reward, 0.0019 ± 0.0009 replay/s; unequal reward, 0.0121 ± 0.004 replay/s; two-sample $t$-test, $t(420) = 3.235$, $p = 0.0013$; familiar saline: equal reward, 0.0033 ± 0.0012 replay/s; unequal reward, 0.0128 ± 0.0025 replay/s; two-sample $t$-test, $t(907) = 3.822$, $p = 0.0001$; familiar CNO: equal reward, 0.0033 ± 0.001 replay/s; unequal reward, 0.0139 ± 0.0022 replay/s; two-sample $t$-test, $t(1153) = 5.234$, $p < 10^{-6}$). The rate of reverse replay was not significantly increased at the Unch. end with unequal rewards in any of these conditions (all $p > 0.05$). This led to a bias for reverse replay to preferentially occur at the Incr. end when rewards were unequal (*Figure 4D, E*). In control rats, reward changes caused similar changes to the balance of reverse replay (*Figure 4H, I*), with a significantly larger swing in reverse replay bias in novel sessions (mixed-effects model with drug, novelty, and replay directionality: novelty × directionality, $z = 2.18$, $p < 0.05$; all other terms, n.s.). The lack of effect of reward in familiar sessions in control rats (*Figure 4I*) was likely due to a dearth of replay in these sessions, with mean reverse replay rates at both reward ends under 0.008 replays/s during Epoch 2, compared to greater than 0.013 replays/s at the Incr. end in Epoch 2 in novel sessions. Reward changes caused no consistent effects in the rates of forward replay (*Figure 4F, G and J, K*), nor in the rates of replay that began far from the animal (non-local replay, *Figure 4—figure supplement 3*).

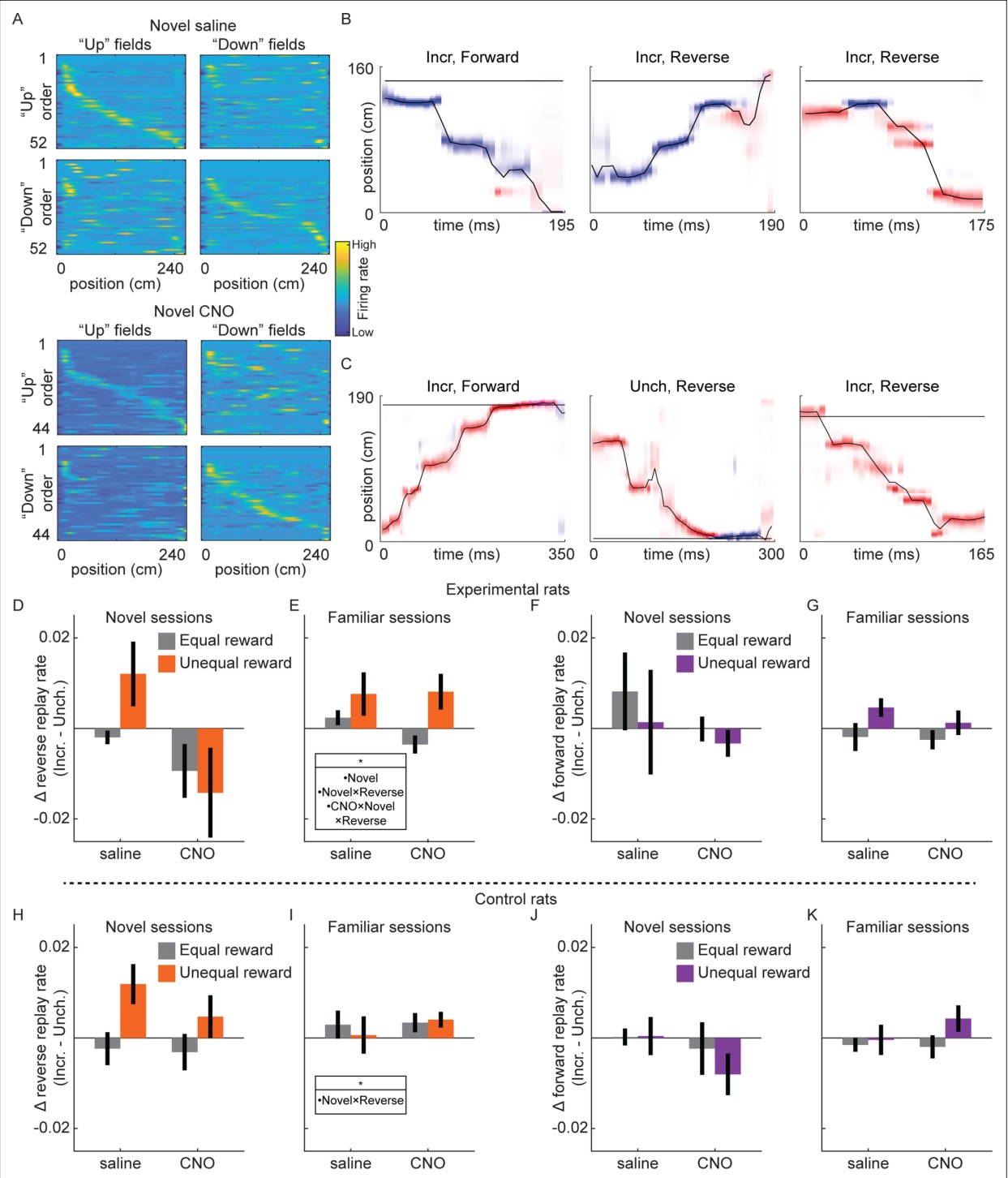

**Figure 4.** Replay recruitment by reward change in novel sessions requires ventral tegmental area (VTA) signaling. (**A**) Place cells exhibit directional place fields on the linear track. Fields calculated from movement in a particular direction ('right' fields and 'left' fields), ordered based on field center location in either running direction ('right' and 'left' orders). Color denotes z-scored firing rate, from high (yellow) to low (blue). Example saline session and CNO session from Experimental rat 3. See also *Figure 4—figure supplements 1 and 2*. (**B**) Three example replays from Epoch 2 of a novel saline session from Experimental rat 3. Red, posterior in upwards map; blue, posterior in downwards map. Title indicates reward end (Incr., Unch.) and replay direction (Reverse, Forward). The horizontal black line indicates rat position. (**C**) Three example replays from Epoch 2 of a novel CNO session from Experimental rat 3, as in (**B**). (**D**) The difference in rate of reverse replay at each end (Incr. – Unch.) in novel sessions in experimental rats. Error bars, standard error of the mean. Reward condition is indicated by color (equal reward, Epoch 1 and 3, gray; unequal reward, Epoch 2, orange), and drug condition is indicated on the x-axis. The difference in replay rate between equal and unequal reward conditions was assessed with a mixed-effects linear model with drug,

*Figure 4 continued on next page*

*Figure 4 continued*

novelty, and replay directionality, and animal-specific intercepts as random effect: drug × novelty × directionality, $z = -2.27$, $p < 0.05$; novelty, $z = -2.01$, $p < 0.05$; novelty × directionality, $z = 2.15$, $p < 0.05$; all other terms, n.s. (**E**) Same as (**D**), but for familiar sessions. (**F**) Same as (**D**), but for forward replay. (**G**) Same as (**F**), but for familiar sessions. (**H**) Same as (**D**), but for control rats. The difference in replay rate between equal and unequal reward conditions was assessed with a mixed-effects linear model with drug, novelty, and replay directionality, and animal-specific intercepts as random effect: novelty × directionality, $z = 2.18$, $p < 0.05$; all other terms, n.s. (**I**) Same as (**H**), but for familiar sessions. (**J**) Same as (**H**), but for forward replay. (**K**) Same as (**J**), but for familiar sessions.

The online version of this article includes the following figure supplement(s) for figure 4:

**Figure supplement 1.** Effect of novelty and ventral tegmental area (VTA) inactivation on place cell properties.

**Figure supplement 2.** Run decoding accuracy in replay analysis sessions.

**Figure supplement 3.** Non-local replay was unaffected by experimental manipulations.

Conversely, in novel CNO sessions in experimental rats, reverse replay rate failed to be biased toward the larger reward location (*Figure 4D*). With unequal reward, the rate of reverse replay did increase at the Incr. end (equal reward, $0.0053 \pm 0.0016$ replay/s; unequal reward, $0.0116 \pm 0.0029$ replay/s; two-sample *t*-test, $t(332) = 2.043$, $p < 0.05$), but also increased somewhat at the Unch. end (equal reward, $0.0173 \pm 0.004$ replay/s; unequal reward, $0.0249 \pm 0.0068$; two-sample *t*-test, $t(319) = 1.019$, n.s.), leading to no consistent change in the difference at the two reward ends. This effect of VTA inactivation on the bias of replay between the reward ends when reward contingencies changed was specific to novel sessions and reverse replay (mixed-effects model with drug, novelty, and replay directionality: drug × novelty × directionality, $z = -2.27$, $p < 0.05$; novelty, $z = -2.01$, $p < 0.05$; novelty × directionality, $z = 2.15$, $p < 0.05$; all other terms, n.s). VTA dopamine signaling was therefore required to direct reward-related changes in reverse replay, specifically in novel environments.

## Discussion

Here, we demonstrated a critical role for VTA dopamine signaling in driving hippocampal SWR and reverse replay selectively to locations with increased reward. Surprisingly, we found this was true only in novel environments, with only modest effects of VTA inactivation on SWR rates and no discernible effect on replay rates in environments that had been explored several times before. We additionally recorded activity in a modified task that allowed us to differentiate SWR rate modulation by value and RPE. While SWR rate was modulated by RPE, VTA inactivation had little effect on this RPE-like modulation, suggesting that at least in familiar environments normal VTA dopamine signaling is dispensable for this reward-related hippocampal activity.

Our work has several limitations. First, we did not directly measure the suppression of VTA neurons after CNO injection. Previous work in other brain areas found hM4Di activation suppressed firing rates to around 60% of baseline (*Mahler et al., 2014*; *Chang et al., 2016*), in addition to diminished synaptic transmission even when spikes occurred (*Stachniak et al., 2014*). Combined with the incomplete expression of hM4Di in TH-positive neurons in our animals, we expect VTA activity was significantly but not completely suppressed. Because our results depend only on any degree of blunting differences in dopamine release at different reward locations, rather than the total absence of dopamine signaling, measuring the magnitude of suppression was not essential for our conclusions. Second, we chose simple behavioral tasks specifically to isolate the effect of reward changes per se on hippocampal replay, distinct from learning or performance effects. In more complex tasks, replay overrepresents high value locations (*Pfeiffer and Foster, 2013*; *Widloski and Foster, 2022*), which may aid in consolidating memory of and planning routes to those locations. In our hands, VTA inactivation reduced the SWR rate difference between high and low value locations in novel environments by about 2/3 (*Figure 2D*), which would greatly reduce any effect of reward-biased replay in learning more complex environments or tasks. Finally, we found no effect of VTA inactivation on SWR rates in familiar environments even with large and frequent reward changes (*Figure 3*), but replicating this result with more animals and probing the interaction between volatile reward changes and environmental novelty are important future directions.

Why is VTA inactivation particularly disruptive during novel experiences? Dopamine neuron firing rates are elevated in novel environments (*McNamara et al., 2014*; *Takeuchi et al., 2016*). More specifically, early in experience dopamine neuron activity ramps while mice run toward both larger and

smaller rewards and this ramping activity declines over experience until modest ramps persist only toward the larger reward (*Guru et al., 2020*). Activation of VTA projections to dorsal CA1 improves retention of spatial learning of a novel maze configuration, while also promoting replay-related reactivation (*McNamara et al., 2014*), while inactivation of VTA causes an increase in spatial working memory-related errors in novel, but not familiar, environments (*Martig et al., 2009*). The results presented here support the hypothesis that VTA is critically involved in learning in new environments, as its inactivation prevents the selective recruitment of replay-associated planning or memory consolidation mechanisms to high value locations.

VTA is not the sole source of dopamine release in hippocampus, with recent work demonstrating LC axons likely provide the bulk of dopamine to dorsal CA1 and can be necessary for novelty-related spatial learning (*Kempadoo et al., 2016*; *Takeuchi et al., 2016*). LC axons in CA1 are active in locations immediately preceding a new reward location in a familiar environment but not in a novel one, despite similar behavior in both cases indicating mice had learned the reward locations (*Kaufman et al., 2020*). This result, coupled with findings that LC neurons are modulated by reward-predicting stimuli similarly to substantia nigra dopamine neurons (*Bouret et al., 2012*; *Bouret and Richmond, 2015*), suggests LC activity can convey reward-related information and thereby compensate for VTA inactivation, but only in familiar environments where it is not signaling more general novelty. Altogether, our work adds to the body of evidence that VTA can directly or indirectly mediate hippocampal plasticity and spatial learning and memory (*Gasbarri et al., 1996*; *McNamara et al., 2014*; *Rosen et al., 2015*; *Rossato et al., 2009*), and suggests an intriguing distinction between the function of VTA and LC dopamine release in hippocampus (*Duszkiewicz et al., 2019*).

These results also support the hypothesis that reverse replay is intimately involved in reward learning (*Ambrose et al., 2016*; *Foster and Wilson, 2006*; *Mattar and Daw, 2018*). By activating representations associated with the location of the current reward and then progressing sequentially to earlier positions that preceded reward, reverse replay may provide a neural eligibility trace by which spatial positions can be associated with their proximity to reward (*Foster and Wilson, 2006*; *Sutton and Barto, 1998*). Dopamine release at reward detection and consumption would then couple a temporal gradient of dopamine concentration with the temporally extended, reverse sequential activation of states that led to that reward. Indeed, VTA dopamine neurons are activated when SWR and replay occur during a spatial working memory task, but not during subsequent sleep (*Gomperts et al., 2015*), indicating close coordination specifically during reward learning. CA1, VTA, and medial prefrontal cortex neurons are jointly coupled via oscillatory mechanisms during spatial working memory (*Fujisawa and Buzsáki, 2011*), suggesting downstream targets of both replay (*Berners-Lee et al., 2021*) and VTA dopamine neurons (*Lammel et al., 2008*) may receive temporally precise conjunctive input from them. We expect future work aimed at untangling under what conditions VTA and replay influence each other and coordinate to provide downstream areas with sequential activity in the presence of dopamine to be particularly fruitful in understanding how reward drives spatial learning.

## Methods

### Lead contact

Further information and requests for resources and reagents should be directed to and will be fulfilled by the Lead Contact, David Foster (davidfoster@berkeley.edu).

### Materials availability

This study did not generate any unique reagents.

### Experimental model and subject details

All experimental procedures were performed in accordance with the University of California Berkeley Animal Care and Use Committee and US National Institutes of Health guidelines. A total of twelve adult male Sprague Dawley TH-cre knock-in rats (inotiv, HsdSage:SD-TH[em1(IRES-Cre)Sage], age 3–10 months, 300–750 g) were used in this experiment, of which seven contributed data to the present report. Two were excluded from further analysis due to lack of virus expression evident in post-mortem immunohistochemistry, two were excluded due to faulty recording hardware, and one was excluded due to non-performance of behavioral tasks. Animals were housed on a standard, non-inverted 12 hr light

cycle. Rats were pair-housed with littermates prior to the start of experiments, after which they were single-housed.

## Behavioral pre-training

Adult male Sprague Dawley TH-cre knock-in rats were fed ad lib and handled daily prior to experimental training. They were then food restricted to 85–90% of baseline weight and trained to collect liquid chocolate reward (0.1 ml) from each end of a single linear track (200 cm length) for at least 15 sessions. Three to six other linear tracks were present in the room during this pre-training, with positions constant for the duration of experiments with each animal.

## Surgical procedures

Each rat underwent virus injection and drive implantation in one surgery (Control rats 1 and 2) or two surgeries spaced 4–20 days apart (Experimental rats 1–4, Control rat 3).

### Virus injection

All virus was obtained from the Stanford Gene Vector and Virus Core under material transfer agreement with the laboratory of Karl Deisseroth. Experimental rats were injected with AAV-DJ-EF1a-DIO-hM4D(Gi)-mCherry (GVVC-AAV-129) and control rats were injected with AAV-DJ-EF1a-DIO-mCherry (GVVC-AAV-14), with 1 μl of virus delivered stereotactically to VTA in each hemisphere (–5.6 mm posterior, ±0.7 mm lateral, and –8 mm ventral, all from bregma and skull surface). Data collection began 4 weeks after virus injection to allow for expression.

### Recording drive implantation

Each rat was implanted with a recording microdrive, targeting bilateral (32 tetrodes, ~40 g, $n$ = 4 rats) or unilateral (20 tetrodes, ~35 g, $n$ = 2 rats; 6 tetrodes, ~20 g, $n$ = 1 rat) dorsal CA1. Each tetrode bundle of four platinum iridium wires (Neuralynx) was independently adjustable and electroplated with gold to an impedance of 150–300 kΩ. Tetrodes were advanced over the course of 1–3 weeks to the pyramidal cell layer. Rats were reintroduced to the pre-training linear track after several days of post-surgical recovery.

## Tissue processing and immunohistochemistry

Eight weeks after virus injection, rats were deeply anesthetized with isoflurane and transcardially perfused with phosphate-buffered saline (PBS) and then 4% paraformaldehyde (PFA) in PBS. Brains were stored in 4% PFA for >24 hr, then 30% sucrose dissolved in PBS for >7 days for cryoprotection. 20–40 μm sections were made in a cryostat and mounted on slides. For TH staining, all steps were performed at room temperature in a dark container on a slow orbital shaker. Sections were rinsed three times for 10 min each in PBS, then incubated for 2 hr in blocker (3% normal donkey serum and 0.3% Triton X in PBS). Sections were then kept for 16–20 hr in blocking buffer with primary antibody (1:200, rabbit α-TH, EMD Millipore 657012, or sheep α-TH, Abcam ab113). After three 10 min washes in PBS, sections were incubated with secondary antibody in blocking buffer for 2 hr (1:200, Alexa Fluor 488-conjugated α-rabbit, Invitrogen Thermo Fisher R37118, or Alexa Fluor 488-conjugated α-sheep, Abcam ab150177). Imaging was performed at the Biological Imaging Facility at the University of California Berkeley using a Zeiss AxioImager M2.

## Drug delivery

At least 10 min prior to beginning a recording session (individual animal means of 12–17.4 min, except for Experimental rat 1, with an average of 4 min before recording session), rats were injected intraperitoneally with saline or 1–4 mg/kg clozapine $N$-oxide (CNO) solution (2–4 mg/ml in diH$_2$O with 50–100 μl dimethyl sulfoxide). In Experiment 1 (described below), 2 mg/kg CNO was delivered in all control animal CNO sessions and in most CNO sessions (22/30 sessions), including all novel track CNO sessions, in three out of four experimental animals. Experimental rat 3 received 1 mg/kg CNO in all sessions, motivated by our desire to have enough low dosage data in one animal in case 2 mg/kg CNO had an effect in control animals (*Gomez et al., 2017*). However, this was not the case, and we saw no difference between CNO doses, consistent with some previous reports in the literature (*Mahler et al., 2014*; *Vazey and Aston-Jones, 2014*), and therefore combined all data. In Experiment

2, animals received 1 mg/kg (*n* = 6 sessions), 2 mg/kg (*n* = 14 sessions), or 4 mg/kg (*n* = 5 sessions), with the higher dose included to probe whether it could reveal a role for VTA in the absence of novelty. As in Experiment 1, we saw little difference between doses and combined all data. One to four sessions (mean ± SD, 1.5 ± 0.65) were completed each day, with at least 1.5 hr between injections. Altogether, we were guided in the injection timing and drug doses by previous work in the field (reviewed in *Smith et al., 2016*), with drug delivery 0–30 min before behavioral testing (*Kane et al., 2017*), significant neuronal suppression occurring within 5–10 min and returning to baseline 70 min post-injection (*Chang et al., 2016*; all recording sessions were completed by 60 min post injection), and CNO doses between 1 and 10 mg/kg (*Smith et al., 2016*).

To prevent the possibility of carry over effects of CNO, saline sessions never followed CNO sessions in the same day (except for 3 recording days in Experimental rat 3, when CNO preceded saline sessions by >4 hr). Besides this restriction, any other ordering of saline and CNO sessions was permitted. In days with two sessions, for example, saline -> saline, saline -> CNO, and CNO -> CNO all occurred. Nevertheless, we attempted to keep the drug X session number of the day close to balanced. In particular, in Experiment 1 novel sessions (described below), each animal had equal numbers of saline and CNO first and second sessions of the day.

## Task design

In Experiment 1, animals progressed through three epochs. In Epoch 1, animals collected 0.1 ml rewards from each end for 10–20 laps. Then, unsignaled to the rat, the session entered Epoch 2, where reward at one end (Incr. end) was increased to 0.4 ml while the other (Unch. end) remained at 0.1 ml. The assignment of track ends to be Incr. and Unch. randomly varied session to session. After 10–20 laps in Epoch 2, the reward changed again unsignaled to the rat, with both reward ends again delivering 0.1 ml in Epoch 3. Rats completed up to 20 laps in Epoch 3, before being removed and placed back into a rest box. This same task was repeated on distinct linear tracks that varied based on position in the room, material of construction, color, length, orientation, and reward well size and position. Sessions were classified as either 'novel' (the first or second experience on a particular linear track) or 'familiar' (third or later experience on a specific track). The track used for pre-training was used first for both saline and CNO sessions. Then, each novel track was used for two to six sessions, with all sessions on a given track consisting of only saline or CNO (excluding one track each in Experimental rats 2 and 3 that had both saline and CNO sessions). The assignment of saline or CNO to each novel track was varied across rats.

In Experiment 2, reward at the stable end (0.2 ml) remained fixed throughout the session, while at the volatile end it varied pseudorandomly every lap between 0 and 0.8 ml (mean 0.37 ml; blocks of 20 laps were comprised of 3 laps × 0 ml, 4 laps × 0.1 ml, 3 laps × 0.2 ml, 4 laps × 0.4 ml, and 6 laps × 0.8 ml). Rats were allowed to continue running until sated. Which track end was assigned to be stable and volatile varied randomly session by session. Each rat performed saline and CNO sessions of this task on the same linear track. The linear track was initially novel (except for in Experimental rat 3). However, one to two saline sessions preceded the first CNO session, rendering it familiar for almost all CNO sessions and most saline sessions. In each rat, all Experiment 2 sessions were completed after all Experiment 1 sessions.

## Data acquisition

Rat position was monitored at 30 frames/s using overhead camera and LEDs on the recording drive, then tracked using automated software (Spike Gadgets). Two-dimensional position and velocity were smoothed using a 7-bin median average, followed by a 5-bin Gaussian filter. Linearized position was then used for further analysis. Neural data was collected using a 128-channel wireless HH128 headstage (Spike Gadgets). LFP was sampled at 30 kHz and spikes extracted based on threshold crossing of 40–60 µV (Trodes software, Spike Gadgets). Individual units were differentiated based on manual clustering of spike waveform peak amplitudes using custom software (xclust2, M. A. Wilson, MIT).

## Behavioral analysis

Reward end visits were defined as periods when the rat was within 10 cm of the end of the track (approximately ~3–5 cm from the reward well, depending on the track). When analyzing visit durations, we excluded a small number of outliers (<15 total across all sessions) that were longer than 60 s.

## LFP analysis

For each session, two to five tetrodes with visible SWR were selected for SWR analysis. LFP from one channel from each tetrode was band-pass filtered between 150 and 250 Hz, then the smoothed (Gaussian kernel, 12 ms SD), absolute value of the Hilbert transform was averaged across tetrodes. For detecting SWR, we examined periods when the rat position was within 10 cm of the reward wells with velocity ≤8 cm/s. SWR were classified as local peaks when the average ripple power exceeded 4 SD above the mean, with start and end points defined as the time ripple power reached the mean before and after the peak, with a minimum start to end duration of 150 ms and maximum of 1 s. SWR rate for each reward end visit was then calculated as the number of SWR detected divided by the total duration of rat velocity ≤8 cm/s during that end visit. During Experiment 1, all analyses considered SWR rate only during the first 10 s of each end visit to isolate the reward consumption-related period and exclude occasional longer task-disengaged resting periods. Results shown in *Figure 2* were similar if we extended this to examine up to the first 20 s of each reward visit. In Experiment 2, we included the first 20 s of each end visit to allow for the longer consumption time required for 0.8 ml.

In Experiment 2, we defined mean-subtracted SWR rate:

$$\text{Mean} - \text{subtracted rate}\,(x, y) = \text{rate}\,(x, y) - \text{rate}\,(x)$$

where *x* and *y* are current and previous volatile volumes, respectively.

## Single-unit analysis

Place fields were calculated for each neuron based on spiking activity while the rat velocity exceeded 8 cm/s. Position was binned into 2 cm bins and directional place fields were calculated as the histogram of spike counts in each position bin (smoothed with Gaussian kernel, 4 cm SD) normalized by the animal's occupancy in each bin, separately for periods when the rat was moving in each direction on the linear track (e.g., left and right). We calculated several properties of place fields to determine whether novelty or VTA inactivation affected them. The map direction correlation was defined as the Pearson correlation between the place field calculated in each running direction, such that a value of 1 indicates perfectly reliable firing dependent only on position, not running direction. The lap to session correlation was defined for each neuron only for the running direction with higher max firing rate. For that running direction, a place field was calculated independently for each lap, smoothed (Gaussian kernel, 4 cm SD), correlated to the directional field calculated from the entire session, and the average correlation coefficient taken across all laps.

## Replay analysis

Because decoding quality relies on sufficient spiking and SWR and population spike density events do not always co-occur, our replay analysis used population spiking to determine candidate events. Candidate replay events were determined based on population activity while the rat was not running (velocity ≤8 cm/s) and near the reward wells (10 cm away at most). Population spike density was binned into 1 ms bins and smoothed (Gaussian kernel, 20 ms SD) and candidate events defined as local peaks when the population rate exceeded the mean by 3 SD and that lasted at least 50 ms, with start and end defined as the nearest times the rate crossed the mean before and after the peak. A memory-less Bayesian decoding algorithm was used to classify both position and running direction during candidate events, as in previous work (*Ambrose et al., 2016*). Replay position in each running direction was estimated in time windows of 40 ms, beginning at the start of the event, and advancing in 5 ms steps. The start/end of putative trajectories within a candidate event were determined by removing bins at either end of the event that contained zero spikes or had a position difference >50% of the track length from the next/previous window. Candidate events with a remaining length of at least five time bins, an absolute weighted correlation (*Wu and Foster, 2014*) exceeding 0.5, and at least 55% of the posterior probability in one of the running directions (*Ambrose et al., 2016*) were classified as replay. Replays were classified as forward or reverse by comparing the direction of replay movement across the track with the running direction map containing the majority of the posterior probability: if they matched (e.g., the replay moved upward and used upward fields), the replay was classified as forward, and otherwise it was classified as reverse.

Only sessions with sufficiently accurate behavioral position decoding accuracy during run were included. Bayesian decoding using the directional place fields was applied to 250 ms non-overlapping

windows covering the entirety of each session. For all time bins with mean animal velocity >20 cm/s, position >20 cm from reward wells, and >5 total spikes from any neurons, actual and decoded position and running direction were compared, yielding a position decoding error (distance in cm) and direction decoding match (same or different). Sessions with mean decoding error >35 cm or direction match <60% were excluded, which totaled 4/64 sessions in experimental rats and 12/72 sessions in control rats.

## Statistics

A mixed-effects Poisson GLM was used to test which experimental variables affected SWR rate using the Matlab *fitglme* function (Mathworks), similarly to previous work (***Ambrose et al., 2016***). Animal ID was modeled as a random effect, allowing baseline SWR rate to vary across rats. The full model was as follows:

SWR rate = exp[b0 + b1 × (Incr. reward end) + b2 × (Epoch 2) + b3 × (CNO) + b4 × (Incr. reward end) × (Epoch 2) + b5 × (Incr. reward end) × (CNO) + b6 × (Epoch 2) × (CNO) + b7 × (Incr. reward end) × (Epoch 2) × (CNO)].

where 'Incr. reward end' is a dummy variable indicating the rat is at the Incr. reward end, 'Epoch 2' is a dummy variable indicating the visit is occurring in Epoch 2, and 'CNO' is a dummy variable indicating it is a session with CNO injected. The coefficient for each term corresponds to the log multiplicative change in SWR rate from the reference condition (animal-specific rate at Unch. end, not in Epoch 2, of a saline session). The offset term log(duration) for each stopping period accounted for the total time the rat was stationary at each reward well visit, so the model fit SWR rate, rather than SWR count. Experimental and control rats were fit separately with this model, as were familiar and novel sessions.

Bootstrapping was used to assess the effect of drug on SWR rate in each experimental condition. For each combination of novelty, reward end, and epoch, drug identity was shuffled 5000 times, generating a distribution of the chance difference between the SWR rate in saline versus CNO sessions. p-values were determined using one-tailed tests, under the hypothesis that CNO would cause lower SWR rates at the Incr. end and higher SWR rates at the Unch. end when compared to saline.

In analyses used to assess the effect of various experimental variables on behavioral and neural measurements, the following variables were consistently defined: 'animal group' was a dummy variable indicating experimental rats, 'drug' was a dummy variable indicating CNO, 'novelty' was a dummy variable indicating novel session, 'epoch' was a categorical variable indicating epoch number, 'reward end' was a dummy variable indicating Incr. reward end, 'RPE sign' was a dummy variable indicating a positive RPE, and 'previous volume' was a categorical variable indicating the volatile volume at the previous visit.

## Acknowledgements

We thank Stanford Gene Vector and Virus Core and Karl Deisseroth for viral constructs and the Biological Imaging Facility at University of California, Berkeley for assistance with tissue imaging. This work was supported by NIH grants NS113557 and MH103325. Animal use conformed to NIH guidelines and was approved by the UC Berkeley Animal Care and Use Committee.

## Additional information

### Funding

| Funder | Grant reference number | Author |
|--------|------------------------|--------|
| National Institute of Neurological Disorders and Stroke | NS113557 | David J Foster |
| National Institute of Mental Health | MH103325 | David J Foster |

| Funder | Grant reference number | Author |
|---|---|---|

The funders had no role in study design, data collection, and interpretation, or the decision to submit the work for publication.

## Author contributions

Matthew R Kleinman, Conceptualization, Formal analysis, Investigation, Methodology, Writing – original draft, Writing – review and editing; David J Foster, Conceptualization, Supervision, Funding acquisition, Methodology, Writing – original draft, Writing – review and editing

## Author ORCIDs

Matthew R Kleinman (iD) https://orcid.org/0009-0002-0221-5577

## Ethics

All experimental procedures were in accordance with the University of California Berkeley Animal Care and Use Committee (protocol AUP-2020-01-12870-1) and US National Institutes of Health guidelines.

Reviewer #1 (Public review): https://doi.org/10.7554/eLife.99678.3.sa1
Reviewer #2 (Public review): https://doi.org/10.7554/eLife.99678.3.sa2
Reviewer #3 (Public review): https://doi.org/10.7554/eLife.99678.3.sa3
Author response https://doi.org/10.7554/eLife.99678.3.sa4

# Additional files

## Supplementary files

MDAR checklist

## Data availability

Custom analysis code has been deposited at Zenodo and is publicly available at DOI https://doi.org/10.5281/zenodo.10368995. All data has been deposited at figshare and is publicly available at DOI 10.6084/m9.figshare.28544036.

The following datasets were generated:

| Author(s) | Year | Dataset title | Dataset URL | Database and Identifier |
|---|---|---|---|---|
| Kleinman MR, Foster DJ | 2025 | KleinmanFoster_2025_eLife | https://doi.org/10.6084/m9.figshare.28544036 | figshare, 10.6084/m9.figshare.28544036 |

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
