## [Editor Report · eLife Assessment]

This work provides a **valuable** contribution to our understanding of the neurobiological mechanisms underlying spatial memory and learning, suggesting that dopamine plays a pivotal role in linking reward context and novelty to memory consolidation processes. The evidence presented to support the main conclusions is **solid**, although reviewers felt that the strength of evidence could have been further strengthened by more rigorous histological verification of the experimental conditions and the complexity of the experimental manipulations, increased sample sizes, and a more consistent approach to experimental dosing and timing, which will be crucial for confirming the reproducibility and reliability of the observed effects.

---

## [Referee Report · Reviewer #1 (Public review)]

This manuscript by Kleinman & Foster investigates the dependence of hippocampal replay on VTA activity. They recorded neural activity from the dorsal CA1 region of the hippocampus while chemogenetically silencing VTA dopamine neurons as rats completed laps on a linear track with reward delivery at each end. Reward amount changed across task epochs within a session on one end of the track. The authors report that VTA activity is necessary for an increase in sharp-wave rate to remain localized to the feeder that undergoes a change in reward magnitude, an effect that was especially pronounced in a novel environment. They follow up on this result with a second experiment in which reward magnitude varies unpredictably at one end of the linear track and report that changes in sharp-wave rate at the variable location reflect both the amount of reward rats just received there, in addition to a smaller modulation that is reminiscent of reward prediction error coding, in which the previous reward rats received at the variable location affects the magnitude of the subsequent change in sharp-wave rate that occurs on the present visit.

This work is technically innovative, combining neural recordings with chemogenetic inactivation. The question of how VTA activity affects replay in the hippocampus is interesting and important given that much of the work implicating hippocampal replay in memory consolidation and planning comes from reward-motivated behavioral tasks.

Comments on revisions:

Overall, I think the authors have done everything they could to address reviewer concerns, short of collecting more data. The more consistent statistical approach makes the paper easier to read and follow. It's helpful to have more details/rationale for the variability in CNO dose and timing. I think some of the results are still not fully convincing, especially the reward volatility experiment (which the authors also note requires additional validation). Given the small number of rats, the small effect sizes, and the complexity of the experimental manipulations, I still have concerns about whether these effects would hold with larger groups sizes.

---

## [Referee Report · Reviewer #2 (Public review)]

(1) Summary

Kleinman and Foster's study investigates the role of dopamine signaling in the ventral tegmental area (VTA) on hippocampal replay and sharp-wave ripples (SWR) in rats exposed to changes in reward magnitude and environmental novelty. The authors utilize chemogenetic silencing techniques to modulate dopamine neuron activity in the VTA while conducting simultaneous electrophysiological recordings from the hippocampal CA1 region. Their findings suggest that VTA dopamine signaling is critical for modulating hippocampal replay in response to changes in reward context and novelty, with specific disruptions observed in replay dynamics when VTA is inhibited, particularly in novel environments.

(2) Strengths

The research addresses a significant gap in our understanding of the neurobiological underpinnings of memory and spatial learning, highlighting the importance of dopamine-mediated processes. The methodological approach is robust, combining chemogenetic silencing with precise electrophysiological measurements, which allows for a detailed examination of the neural circuits involved. The study provides important insights into how hippocampal replay and SWR are influenced by reward prediction errors, as well as the role of dopamine in these processes. Specifically, the authors note that VTA silencing unexpectedly did not prevent increases in ripple activities where reward was increased, but induced significant aberrant increases in environments where reward levels were unchanged, highlighting a novel dependency of hippocampal replay on dopamine and a VTA-independent reward prediction error signal in familiar environments. These findings are critical for understanding the consolidation of episodic memory and the neural basis of learning.

(3) Weaknesses

Despite the strengths in methodology and conceptual framework, the study has several weaknesses that could affect the interpretation of the results. There is a need for more rigorous histological validation to confirm the extent and specificity of viral expression (from all animals ideally), which is crucial for ensuring the accuracy of the findings. Variability in the dosing and timing of chemogenetic interventions could also lead to inconsistencies in the data, suggesting a need for more standardized experimental protocols.

Comments on revisions:

I commend the authors for their work in addressing my and the other reviewers' comments. I think these changes have improved the paper, and no further changes are absolutely necessary.

---

## [Referee Report · Reviewer #3 (Public review)]

Summary:

The authors of this work are trying to understand the role dopaminergic terminals coming from VTA have on hippocampal mechanisms of memory consolidation, with emphasis on the replay of hippocampal patterns of activity during periods of consummatory behavior in reward locations. Previous work suggested that replay of relevant spatial trajectories supports reward localization and influences behavior.

The authors then tried to separate two conditions that were known to cause an increase in replay activity - spatial novelty encoding and variation of reward magnitude - and evaluate how these changed when VTA dopamine neurons were inactivated by a chemogenetic tool. They found that the rate of reverse replay (trajectory going away from the goal location) is increased with reward only in novel, but not in familiar environments. Overall this suggests that the VTA dopamine signal is critical during learning of novel locations, but not during explorations of already familiar environments.

Strengths:

The inactivation of VTA projections during goal-oriented behavior and in-vivo analysis of patterns of hippocampal activity during both novelty and reward variability. This work adds to the body of evidence that reverse replay constitutes an important mechanism in learning spatial goal locations. Furthermore, this work also points to the role of VTA in reward prediction error with consequences for spatial navigation and consolidation of spatial memories.

The authors addressed very carefully all the points raised during the revision and I am very pleased with the revised manuscript.

---

## [Author Response]

The following is the authors’ response to the original reviews.

**Reviewer #1:**
Chemogenetics validationLittle validation is provided for the chemogenetic manipulations. The authors report that animals were excluded due to lack of expression but do not quantify/document the extent of expression in the animals that were included in the study.

We thank the reviewer for raising this oversight. We have added additional examples of virus expression in sections from included and excluded animals in Figure 1 – Supplement 1. We also added additional comments on the extent of expression we observed in lines 92-95: “Post-experiment histology confirmed overlapping virus expression and TH-positive neurons in putative VTA near the injection site (-5.6 mm AP from bregma), as well as approximately 0.5 mm anterior and posterior (-5 to -6 mm AP).”

There's no independent verification that VTA was actually inhibited by the chemogenetic manipulation besides the experimental effects of interest.

While we did include animals expressing control virus to control for any effect of CNO administration itself, the reviewer is correct that we did not independently verify VTA neurons were inhibited. We have noted this limitation of the current study on lines 513-522 in the Discussion: “We did not directly measure the suppression of VTA neurons after CNO injection. Previous work in other brain areas found hM4Di activation suppressed firing rates to around 60% of baseline (Mahler et al., 2014; Chang et al., 2015), in addition to diminishing synaptic transmission even when spikes occurred (Stachniak et al., 2014). Combined with the incomplete expression of hM4Di in TH-positive neurons in our animals, we expect VTA activity was significantly but not completely suppressed. Because our results depend only on any degree of blunting differences in dopamine release at different reward locations, rather than the total absence of dopamine signaling, measuring the magnitude of suppression was not essential for our conclusions.”

The authors report a range of CNO doses. What determined the dose that each rat received? Was it constant for an individual rat? If not, how was the dose determined? The authors may wish to examine whether any of their CNO effects were dependent on dose.

The reviewer is completely correct that we omitted sufficient information regarding the dosage of CNO used in each animal and each session. We have included more details in the Methods lines 676-694, detailing both the doses and the rationale.

The authors tested the same animal multiple times per day with relatively little time between recording sessions. Can they be certain that the effect of CNO wore off between sessions? Might successive CNO injections in the same day have impacted neural activity in the VTA differently? Could the chemogenetic manipulation have grown stronger with each successive injection (or maybe weaker due to something like receptor desensitization)? The authors could test statistically whether the effects of CNO that they report do not depend on the number of CNO injections a rat received over a short period of time.

We thank the reviewer for bringing up the question of whether the order of sessions had an influence on the efficacy of CNO in inactivating VTA activity. To address this, we split our dataset in Experiment 1 into two based on what number session of the particular day it was: 1st sessions of the day vs. all subsequent sessions (2nd+ session of the day). Then, we examined the difference in sharp-wave ripple rate between the reward ends in Epoch 2, as in Figure 2D of the manuscript. Though the resulting number of sessions in each split of the dataset is too low to draw strong statistical conclusions, particularly for novel sessions, we see little evidence there is any systematic change in the effect of VTA inactivation as a function of session number in the day. We include this in the revised manuscript as Figure 2 – Supplement 3 and in the Results lines 255-258.

Motivational considerationsIn a similar vein, running multiple sessions per day raises the possibility that rats' motivation was not constant across all data collection time points. The authors could test whether any measures of motivation (laps completed, running speed) changed across the sessions conducted within the same day.

We thank the reviewer for this suggestion. We examined behavioral measures of motivation across sessions conducted within the same day. First, we calculated how many total laps each animal completed each session as a function of the session number of the day. In individual animals, this ranged from -2.8 to 4.1 laps per additional session number (mean 2.01), with an average total laps per session of 43.2 laps. Second, we calculated the median running velocity per session, across both running directions and all epochs, and checked how it varied across session number of the day. Per additional session in the day, this ranged from -3.6 to 8.6 cm/s difference across animals (mean 2.7 cm/s), with an average running velocity of 34.1 cm/s in total. Taken together, while we found little behavioral evidence of strong motivational changes across session, our animals may have been slightly more motivated in later sessions in the day, which also corresponded to later in the light cycle and closer to the dark cycle. We mention this information in Results lines 255-258, related to Figure 2 – Supplement 3.

This is a particularly tricky issue, because my read of the methods is that saline sessions were only conducted as the first session of any recording day, which means there's a session order/time of day and potential motivational confound in comparing saline to CNO sessions.

We have clarified the ordering of CNO and saline sessions in the Methods lines 697-702. Briefly, we avoided running CNO sessions before saline sessions in the same day, but either could be the first session of a day. That is, saline -> saline, saline -> CNO, and CNO -> CNO were all valid orderings. On days with more than two sessions, any number of repeated saline and CNO sessions was permitted, provided that as soon as a CNO session occurred, any subsequent sessions were also CNO.

More generally, we shared this reviewer’s concern about potential confounds between drug and motivation. For novel sessions in Experiment 1, each animal had equal numbers of saline and CNO 1st and 2nd sessions of the day. For familiar sessions, animals had similar counts for 1st sessions of the day (experimental rats: 20 saline, 16 CNO; control rats: 17 saline, 15 CNO) but more CNO 2nd sessions of the day (experimental rats: 5 saline, 13 CNO; control rats: 5 saline, 10 CNO). There were occasionally 3rd or 4th sessions in a given day for some rats, and these were also approximately equal (experimental rat 2, 3rd sessions: 2 each of saline and CNO, 4th session: 1 saline; experimental rat 3 and 4, 3rd sessions: 1 each of saline and CNO; control rat 2, 3rd session: 1 saline).

Statistics, statistical power, and effect sizesThroughout the manuscript, the authors employ a mixture of t-tests, ANOVAs, and mixed-effects models. Only the mixed effects models appropriately account for the fact that all of this data involves repeated measurements from the same subject. The t-tests are frequently doubly inappropriate because they both treat repeated measures as independent and are not corrected for multiple comparisons.

We thank the reviewer for pointing out these issues with our statistical analyses in places. We have made the following improvements:

Figure 1F-I, S1, reward end visit durations: We now use a linear mixed-effects model to analyze the difference in stopping period durations between epochs. For each session, we calculated the mean stopping duration for each reward end in each epoch, then modeled the difference between epochs as a function of drug and novelty, with animal-specific intercepts. For example, related to Figure 1F and also described in the Results, we modeled the stopping duration difference at the Unchanged reward end, Epoch 2 – Epoch 1, and found experimental rats had a significant intercept (Epoch 2 stops shorter than Epoch 1) and the drug × novelty interaction, while control rats had a significant intercept and novelty main effect. The other visit duration analysis shown in Figure 1 – Supplement 1 have similarly been updated.

Figure 2D-E, ripple rate difference between reward ends in Epoch 2: We now use a linear mixed-effects model to analyze the difference between ripple rates at the Incr. and Unch. reward ends in Epoch 2. For each session, we calculated the mean ripple rate at each end in Epoch 2, then modeled the difference as a function of drug and novelty, with animal-specific intercepts. With the full stopping periods, for experimental rats, there was a significant intercept (ripple rate at Incr. greater than Unch.) and the model with drug included performed significantly better than the one without it (AIC_nodrug_ – AIC_full_ = 5.22). Control rats had a significant intercept and effect of novelty (greater difference with novelty), and the model excluding drug terms performed better (AIC_nodrug_ – AIC_full_ = -3.54). Results with the trimmed stopping periods were similar. These analyses are described in Results lines 253-266.

Figure 3D-E, ripple rate as a function of reward history: We now use a mixed-effects model that incorporates animal-specific intercepts. The results remained similar and have been updated in the text and legend.

Figure 4D-K, replay rates as a function of drug, novelty, and directionality: We now use mixed-effects models that incorporate animal-specific intercepts rather than three-way ANOVA. The results remained similar and have been updated in the text and legend.

The number of animals in these studies is on the lower end for this sort of work, raising questions about whether all of these results are statistically reliable and likely to generalize. This is particularly pronounced in the reward volatility experiment, where the number of rats in the experimental group is halved to just two. The results of this experiment are potentially very exciting, but the sample size makes this feel more like pilot data than a finished product.

We have added additional emphasis in the text that the experimental group results of CNO inactivation in the volatile reward task should be confirmed with future work, in Discussion line 529-533. Because these experiments were performed on familiar tracks, we see them as corroborating/complementing the results from Experiment 1. Although the analysis assumes VTA inactivation had no effect, our pooling of all Experiment 2 data to display in Figure 3 – Supplement 2 maximized our ability to analyze the effects of volatile reward deliveries on sharp-wave ripple rates, lending further support to the main results shown in Figure 3.

The effect sizes of the various manipulations appear to be relatively modest, and I wonder if the authors could help readers by contextualizing the magnitude of these results further. For instance, when VTA inactivation increases mis-localization of SWRs to the unchanged end of the track, roughly how many misplaced sharp-waves are occurring within a session, and what would their consequence be? On this particular behavioral task, it's not clear that the animals are doing worse in any way despite the mislocalization of sharp-waves. And it seems like the absolute number of extra sharp-waves that occur in some of these conditions would be quite small over the course of a session, so it would be helpful if the authors could speculate on how these differences might translate to meaningful changes in processes like consolidation, for instance.

We thank the reviewer for this helpful suggestion to give some context to the difference in sharp-wave ripple numbers and the functional consequence of these changes. We agree completely that this task is almost certainly too simple for animals to show any performance deficit from these changes. We chose this precisely so we could examine the consequences of VTA inactivation to the sharp-wave ripple response to reward changes per se, without any confound of performance or memory changes that could also conceivably alter sharp-wave ripples. We have added both more context about the magnitude and consequence of these sharp-wave ripple changes as well as comments about the choice of this particular task (Discussion lines 522-529).

How directly is reward affecting sharp-wave rate?Changes in reward magnitude on the authors' task cause rats to reallocate how much time they spent at each end. Coincident with this behavioral change, the authors identify changes in the sharp-wave rate, and the assumption is that changing reward is altering the sharp-wave rate. But it also seems possible that by inducing longer pauses, increased reward magnitude is affecting the hippocampal network state and creating an occasion for more sharp-waves to occur. It's possible that any manipulation so altering rats' behavior would similarly affect the sharp-wave rate.For instance, in the volatility experiment, on trials when no reward is given sharp-wave rate looks like it is effectively zero. But this rate is somewhat hard to interpret. If rats hardly stopped moving on trials when no reward was given, and the hippocampus remained in a strong theta network state for the full duration of the rat's visit to the feeder, the lack of sharp-waves might not reflect something about reward processing so much as the fact that the rat's hippocampus didn't have the occasion to emit a sharp-wave. A better way to compute the sharp-wave rate might be to use not the entire visit duration in the denominator, but rather the total amount of time the hippocampus spends in a non-theta state during each visit. Another approach might be to include visit duration as a covariate with reward magnitude in some of the analyses. Increasing reward magnitude seems to increase visit duration, but these probably aren't perfectly correlated, so the authors might gain some leverage by showing that on the rare long visit to a low-reward end sharp-wave rate remains reliably low. This would help exclude the explanation that sharp-wave rate follows increases in reward magnitude simply because longer pauses allow a greater opportunity for the hippocampus to settle into a non-theta state.

We thank the reviewer for these important comments. We have better clarified the analysis of sharp-wave ripple rate in the Results (lines 172-173). To speak to the main concern of the reviewer, we do only consider times during “stopping periods” when the rat is actually stationary. That is, ripple rate for each visit is calculated as (# of ripples / total stationary time), rather than the full duration the rat is at the track end. With respect to including visit duration as a covariate, the Poisson model takes the total stationary time of each visit into account, so that it is effectively predicting the number of events (ripples) per unit of time (seconds) given the particular experimental variables (reward condition, drug condition, etc.). We have added additional clarification of this in the Methods (line 834-836).

The authors seem to acknowledge this issue to some extent, as a few analyses have the moments just after the rat's arrival at a feeder and just before departure trimmed out of consideration. But that assumes these sorts of non-theta states are only occurring at the very beginning and very end of visits when in fact rats might be doing all sorts of other things during visits that could affect the hippocampus network state and the propensity to observe sharp-waves.

We hope that with the clarification provided above, this control analysis helps remove any potential effects of approaching/leaving behavior or differences in movement at the reward end that could alter sharp-wave ripple rates.

Minor issuesThe title/abstract should reflect that only male animals were used in this study.

We have added this important information to the Abstract line 21.

The title refers to hippocampal replay, but for much of the paper the authors are measuring sharp-wave rate and not replay directly, so I would favor a more nuanced title.

We thank the reviewer for this suggestion. In the context of our work, we consider sharp-wave ripples as more-easily-detected markers for the occurrence of replay. Previous work from our lab (Ambrose et al., 2016) showed the effect of reward changes had very similar effects to both sharp-wave ripple rate and replay rate. We try to be explicit about viewing ripples as markers of replay content in both the Introduction and Discussion. Nevertheless, we do also demonstrate the title claim directly – by measuring replay and its spatial localization – therefore we feel comfortable with the title as it is.

Relatedly, the interpretation of the mislocalization of sharp-waves following VTA inactivation suggests that the hippocampus is perhaps representing information inappropriately/incorrectly for consolidation, as the increased rate is observed both for a location that has undergone a change in reward and one that has not. However, the authors are measuring replay rate, not replay content. It's entirely possible that the "mislocalized" replays at the unchanged end are, in fact, replaying information about the changed end of the track. A bit more nuance in the discussion of this effect would be helpful.

While we do show that replay content, in the form of reverse vs. forward replays, is altered with VTA inactivation, we take the reviewers point and completely agree. Especially in the context of the linear track, replays at either end could certainly be updating/consolidating information about both ends. We would argue our results suggest VTA is critical to localizing ripples and replay in more complex environments where this is not the case, but this is a hypothesis. We have added clarification and discussion of this point (Discussion lines 522-529).

However, in response to the reviewer’s comment, we have now also examined non-locally-initiated replays specifically to determine whether the increased ripple rate at the Unch. reward end in novel CNO sessions was likely due to more non-local replay, but found no significant increases in non-local replay at either reward end in either drug condition or novelty condition. We have included this result as Figure 4 – Supplement 3, and note it in the Results lines 487-488.

The authors use decoding accuracy during movement to determine which sessions should be included for decoding of replay direction. Details on cross-validation are omitted and would be appreciated. Also, the authors assume that sessions failed to meet inclusion criteria because of ensemble size, but this information is not reported anywhere directly. More info on the ensemble size of included/excluded sessions would be helpful.

We have added additional information about the run decoding procedure and related session inclusion criteria, as well as about recorded ensemble sizes (lines 417-421). Briefly, mean ensemble sizes were significantly smaller for excluded sessions (cell count, mean±sem; included sessions: 26.1±1.1, excluded sessions: 9.5±1.6; two-sample t-test, t(133)=5.3, p<10^-5^). The average field size, defined as the number of spatial bins with greater than 1 hz firing rate, in excluded sessions was also larger (mean±sem, included sessions: 47.7±1.3, excluded sessions: 57.7±5.8; two-sample t-test, t(133)=-2.33, p<0.05), though the difference was less dramatic. Using a mixed effects model to predict position decoding error (as in Figure 4 – Supplement 2A) as a function of drug, novelty, cell count, and mean place field size, in both experimental and control groups cell count and field size were significant predictors: more cells and smaller average field size led to lower error. A similar model that instead predicted the fraction of running bins with correctly decoded running direction (as in Figure 4 – Supplement 2B), in neither group was field size significant, while cell count remained so: more cells led to more bins with running direction correctly classified. We include these analyses in the legend for the figure. With respect to cross validation of run decoding, because both the contribution of spikes in any single time bin to a neuron’s place field is extremely small and because we used run decoding accuracy simply to filter out sessions with poorer decoding, we did not use cross validation here.

For most of the paper, the authors detect sharp-waves using ripple power in the LFP, but for the analysis of replay direction, they use a different detection procedure based on the population firing rate of recorded neurons. Was there a reason for this switch? It's somewhat difficult to compare reported sharpwave/replay rates of the analyses given that different approaches were used.

We have added clarification for this change in detecting candidate events (lines 787-789). Briefly, sharp-wave ripples and spike density events are often but not always overlapping, such that there can be strong ripples with little spiking in the recorded ensemble or weak/absent ripples during vigorous spiking in the recorded ensemble. Because the decoding of replay content relies on spiking, our lab and others often use spike density or population burst events as candidate events. We have confirmed that the main results of Experiment 1 (e.g., Figure 2) remain the same if we use spike density events rather than sharp-wave ripples, but prefer to keep the use of sharp-wave ripples here for better comparison with Experiment 2 and to allow the inclusion of animals and sessions with low cell yield but clear ripples in the LFP.

**Reviewer #2 (Recommendations For The Authors):**
Include additional histological data to confirm the extent of viral spread and precise tetrode placements. Providing detailed figures that clearly illustrate these aspects would strengthen the validity of the neural recordings and the specificity of the chemogenetic silencing.

We thank the reviewer for this suggestion and have added additional information regarding virus expression in Figure 1 – Supplement 1. We also added additional comments on the extent of expression we observed in lines 92-95: “Post-experiment histology confirmed overlapping virus expression and TH-positive neurons in putative VTA near the injection site (-5.6 mm AP from bregma), as well as approximately 0.5 mm anterior and posterior (-5 to -6 mm AP).”

While we do not show histological confirmation of hippocampal recording sites, the presence of sharp-wave ripples with upward deflections, presence of place cells, and recording coordinates and depth typical of dorsal CA1 made us confident in our recording location. We have noted these characteristics of our recordings in lines 128-131 in the Results: “Tetrodes were lowered to the pyramidal cell layer of dCA1, using the presence of sharp-wave ripples with upward deflections in the LFP, recording depth characteristic of dCA1, and spatially-restricted firing of place cells to confirm the recording location.”

Address the variability in CNO dosing and timing before recordings. It is recommended to standardize the dose and ensure a consistent timing interval between CNO administration and the start of recordings to minimize variability in the effects observed across different subjects. Instead of collecting new data, the authors could report the data for each animal, indicating the dose and interval between the injection and the recording.

We have further clarified the CNO dosing and timings in lines 676-702.

In Figure 1F, explicitly state whether the data represent averages across multiple sessions and confirm if these observations are primarily from the initial novel sessions. This clarification will help in accurately interpreting the effects of novelty on the measured neural activities.

We have changed the analyses shown in Figure 1F-I and Figure 1 – Supplement 1 thanks to the suggestions of Reviewer #1, but also more clearly spell out the analysis. Briefly, we average the durations for each condition within session (e.g., take the mean Unch. duration in Epoch 1), then perform the analysis across sessions. These data come from all sessions in Experiment 1, as described in lines 141-147, meaning there are around 2-3 times as many familiar sessions as novel sessions.

Reconsider the reporting of marginal p-values (e.g., p=0.055). If the results are borderline significant, either more data should be collected to robustly demonstrate the effects or a statistical discussion should be included to address the implications of these marginal findings.

We have removed the reporting of marginal p-values.

Ensure that the axes and scales are consistent across similar figures (specifically mentioned for Figure 2A) to prevent misinterpretation of the data. Consider showing the average across all animals in 2A, similar to 2B and 2C.

We have adjusted these axes to be consistent across all panels.

Add a legend to the heatmap in Figure 4A to facilitate understanding of the data presented.

We have added a heatmap to the figure and legend.

Provide a detailed examination and discussion of the apparent contradictions observed in control data, particularly where experimental conditions with saline show increased reverse replay in novel environments, which is absent in familiar sessions. See Figures 4E and 4I.

We thank the reviewer for noting that this feature of our data deserved discussion. We confirmed that the lack of an effect of reward on reverse replay rates in familiar sessions in control rats was due to generally low replay rates in these sessions. Replay rates have been observed to decrease as the familiarity of an environment or behavior increases, and the presence of the reward-related modulation of reverse replay in novel sessions in these animals is consistent with this observation. We now report in the Results lines 458-459 and 485-486 the low replay rates in this group in familiar sessions, and the likelihood that this is preventing any reward-related modulation from being detected.

Include a more detailed analysis of place cell properties, such as firing rates and field sizes, especially in novel environments where VTA inactivation appears to alter spatial coding. Decoding error is lower during CNO administration - does this mean place fields are smaller/more accurate? This analysis could offer deeper insights into the mechanisms by which dopamine influences hippocampal neural representations and memory processes.

We thank the reviewer for this helpful suggestion. We have expanded on our analysis of place field properties and decoding accuracy, describing properties of sessions with good enough decoding to be included compared to those that were excluded (lines 417-421). We also directly tested how decoding quality depended on several factors, including drug condition, novelty, number of cells recorded, and the average place field size of recorded cells (see legend for Figure 4 – Supplement 2). We found a small but significant effect of drug in experimental rats, but larger effects of number of recorded cells and average field size, that were also present in control animals.

Correct the typo on line 722 from "In ANOVA" to "An ANOVA".

We reworded this section and have corrected this error.

**Reviewer #3 (Recommendations For The Authors):**
The manuscript is clear and exciting. As a main criticism, I would have liked to see the effects on ripple duration not just the rate.

We thank the reviewer for this interesting idea. We performed a new analysis, similar to our analysis on SWR rate, probing the effect of our experimental manipulations on SWR duration in experimental rats. We have added the results in Figure 2 – Supplement 4, and note them in the main text lines 195-198: “SWR duration was reduced in novel sessions, consistent with replays becoming longer with increased familiarity (Berners-Lee et al., 2021), as well as in Epoch 2, but was otherwise unaffected by reward or drug (Figure 2 – Supplement 4).”

I have a few other minor comments:(1) I find it a little disturbing and counterintuitive that statistical differences are not always depicted in the figure graphs (for example Figures 2A-E). If the authors don't like to use the traditional *, ** or *** they could either just use one symbol to depict significance or simply depict the actual p values.

We thank the reviewer for this suggestion. We struggled with indicating significance values graphically in an intuitive way for interaction terms in the figures. We now added significance indicators in Figures 1F-I, added the significant model coefficients directly into Figure 2B-C, changed the analysis depicted in Figure 2D-E per Reviewer 1’s suggestions, and added significance indicators where previously missing in Figures 3 and 4.

(2) Related to the point above: in the page 7 legend D and E, it would be advantageous for clarity of the experimental results to also perform post-hoc analyses as depicted in the graphs, rather than just describe the p-value of the 3way ANOVA;

We thank the reviewer for this suggestion. Because the figure includes the mean and standard error of each condition, in addition to the significant effects of the mixed-effects model, we prefer the current format as it makes clearer the statistical tests that were performed while still allowing visual appreciation of differences between specific experimental conditions of interest to the reader.

(3) According to Figure 1H, the duration of the reward visits can go up to 15s (or more). Yet in Figure 2A only the first 10sec were analyzed. While I understand the rationale for using the initial 10 seconds where there is a lot more data, the results of graphs of Figures A to C will not have the same data/rate as Figures D-F where I assume the entire duration of the visit is taken into account.A figure showing what happening to the ripple rate during the visits >10sec would help interpret the results of Figure 2.

We thank the reviewer for these interesting suggestions. We clarify now that all these analyses of Experiment 1 use only the first 10 s of each stopping period in Method line 758-764. However, examining the longer stopping periods is an excellent suggestion, and we re-analyzed the Experiment 1 dataset using up to the first 20 s of each stopping period. The main results (e.g., Figure 2) remain the same:

(1) Related to Figure 2B-C: For experimental rats, a mixed-effects generalized linear model predicting sharp-wave ripple rate as a function of reward end, block, drug, novelty, and interactions, had the following significant terms: drug (p<10^-5^), novelty (p<10^-10^), reward end × block (p<10^-10^), and reward end × block × drug (p<0.05). The same model in control rats had significant terms: reward end (p<0.05), novelty (p<10^-4^), reward end × block (p<10^-10^).

(2) Related to Figure 2D-E: For experimental rats, we used a mixed-effects generalized linear model predicting the difference in sharp-wave ripple rate between the Incr. and Unch. reward ends in Epoch 2 as a function of novelty, drug, and their interaction. Model comparison found the full model performed better than a model removing the drug terms (AIC_nodrug_ – AIC_full_ = 2.94), while a model with only the intercept performed even worse (AIC_intercept_ – AIC_full_ = 13.76). For control rats, model comparison found the full model was equivalent to a model with only the intercept (AICintercept – AICfull = -0.36), with both modestly better than a model with no drug terms (AIC_nodrug_ – AIC_full_ = 1.38).

We have added a remark that results remain the same using this longer time window in Methods line 758-764.